# GSINA: Improving Graph Invariant Learning via Graph Sinkhorn Attention

## Abstract

Graph invariant learning (GIL) has been extensively studied to discover the invariant relationships between graph data and its labels for different graph learning tasks under various distribution shifts. Many recent endeavors of GIL focus on discovering invariant features to improve the generalization performance of graph learning. However, existing methods often have limitations in obtaining invariant features expressive enough in the solution space. In this paper, we analyze the limitations of previous works and briefly summarize the design principles of the invariant feature for GIL as 3 aspects: 1) the sparsity, to filter out the variant features, 2) the softness, for a broader solution space, and 3) the differentiability, for a soundly end-to-end optimization. To meet these principles in one shot, we leverage the Optimal Transport (OT) theory and propose a novel graph attention mechanism: Graph Sinkhorn Attention (GSINA) as a powerful regularization method for GIL tasks, by which we could obtain meaningful differentiable graph invariant features with controllable sparsity and softness. Moreover, GSINA as a general graph representation learning framework could handle GIL tasks of multiple data grain levels. Experiments on both synthetic and real-world datasets validate the superiority of our GSINA, which outperforms the state-of-the-art GIL methods (GSAT, CIGA, EERM) by large margins on graph-level tasks and node-level tasks. The PyTorch source code is provided in supplementary materials and will be publicly available on GitHub.

## 1 Introduction

Graph data is ubiquitous in real-world applications, e.g. social networks (1), supply chain networks (2), and chemical molecules (3). Graph machine learning, especially graph neural networks (GNNs), has shown promising results in various graph-related tasks (4; 5; 6). Despite their success, existing approaches often rely on the I.I.D. assumption, assuming the train and test graph data are drawn from the same distribution. However, distribution shifts, i.e., the mismatches between different data domains widely exist especially for complex graph data. The out-of-distribution (OOD) generalization has become a main obstacle and hot topic in graph representation learning.

In particular, graph invariant learning (GIL), which aims to capture the invariant relationships between graph data and labels for graph OOD generalization, has been extensively studied in various generalization tasks such as graph-level (7; 8; 9; 10; 11; 12) and node-level (13; 14; 15) tasks. GIL can be roughly divided into two research lines, namely explicit representation alignment and invariance optimization (16). The main idea of the explicit representation alignment methods is to align graph representations among multiple environments. These methods are designed to minimize the difference across various environments with the regularization strategies (14; 8; 15). The invariance optimization methods are based on the principle of invariance, which assumes the invariant property inside data or the invariant features under distribution shifts. Many of the invariance optimization methods are aimed at handling graph OOD generalization by discovering graph invariant features (e.g. crucial nodes and edges) under distribution shifts (9; 10; 11). As empirical collateral evidence, crucial graph information usually exists in a few edges and nodes in real-world scenarios. For instance, in the chemical field, key functional groups in a molecule yield a certain property like solubility (17). In the financial risk management field, the risk level of a community is often determined by a few key members (18).

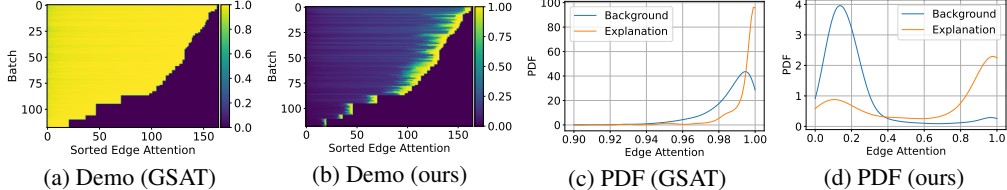

|     |     |     |     |
| --- | --- | --- | --- |
| (a) Demo (GSAT) | (b) Demo (ours) | (c) PDF (GSAT) | (d) PDF (ours) |

Figure 1: Subgraph sparsity of GSAT (10) and our GSINA ($r = 0.3$) on a batch of data in SPMotif ($b = 0.5$) dataset. Fig. 1a, 1b demonstrate the learned edge importances (attention values) by GSAT and our GSINA for each edge of the graphs in the batch, the bottom right black regions are non-edge padding. The X-axis represents edges (sorted by attention values), and the Y-axis represents graphs. Fig. 1c, 1d are the PDF plots of edge attention distributions generated by GSAT and our GSINA for the edges in Background (label-independent) part and Explanation (label-related) part.

The invariance optimization methods (10; 9; 11) have made considerable efforts for the sound generalizability and inherent interpretability of their invariant feature discovery, and there are two mainstream schemes: 1) the *Information Bottleneck (IB)* (19) scheme (20; 10) exploits IB principle to

Table 1: Comparison with SOTA invariance optimization GIL methods.

|  | Sparse | Soft | Differentiable |
| --- | --- | --- | --- |
| GSAT (10) (IB) | No | Yes | Fully |
| CIGA (11) (top-$k$) | Yes | No | Partially |
| Ours | **Yes** | **Yes** | **Fully** |

extract label-relevant graph invariant features with information constraint, 2) the *top-k (Subgraph Selection)* scheme (9; 11) intuitively chooses the top-$k$ most influential edges as the 'invariant subgraph'. Despite the success of those studies, there are still limitations that need to be stated and addressed for the expressiveness of their methods. Firstly, The IB based methods might not proactively guarantee the sparsity of subgraphs (10). As shown in Fig. 1, we provide a demonstration with the latest IB based GIL SOTA: Graph Stochastic Attention (GSAT) (10), which evaluates the importance of each edge by assigning edge attention, and as shown in Fig. 1a, 1c, GSAT lacks sparsity: the edge attention for the Background part and the Explanation part are similar, making it difficult to make a prediction based on the most valuable invariant features, which are supposed to be more distinguishable. Secondly, the *top-k* based methods capture the invariant subgraph in a 'hard' way, i.e. only the top-$k$ part is kept for training and prediction, and the other part is neglected, as only restricted information is utilized, hard subgraph extraction results in a restricted solution space to find the optimal invariant subgraph, and problematically, an ill-posed optimization: the top-$k$ selection operation itself is not differentiable (it does not provide gradients for model backward pass), to make them trainable, these methods are 'partially differentiable' for learning after assigning the differentiable weighting scores output from their subgraph extractors for top-$k$ selection to the extracted subgraph (as the practices in (9; 11)). Moreover, as the top-$k$ selection discards part of the graph structure, these methods could only be used for the tasks of graph level, for a grainer task level, e.g. node level, these methods are not applicable as the graph structure is incomplete (part of the nodes have been discarded), and there is no way to learn complete node representations.

To address the above-mentioned issues of invariance optimization GIL, we first summarize 3 principles for a graph invariant feature extractor: 1) sparsity (as shown in Fig. 1b, 1d) to effectively filter out the variant features, and the invariant subgraph should be sufficiently distinguishable to avoid confusion with the variant part, 2) softness (compared with the 'hard selection' ways) to enlarge the subgraph solution space, to numerically evaluate the graph feature importances, and no graph information omissions, and 3) differentiability (also based on softness) for a soundly end-to-end optimization, and to ensure the invariant subgraph extractor could be learned to generate sparse and soft subgraphs. Then, besides their (9; 10; 11) limitations, their effectiveness also has inspired our work from various aspects, as summarized in Tab. 1, GSAT (10) is soft and differentiable as it is based on graph attention mechanism, DIR (9) and CIGA (11) are sparse as their top-$k$ operations explicitly constrain the proportion of their subgraphs to their input graphs. Additionally inspired by the recent advances in cardinality-constrained combinatorial optimization (21; 22), the top-$k$ problem could be addressed by a series of soft and differentiable iterative numerical calculations of the Optimal Transport (OT) (23) theoretic Sinkhorn algorithm (24), resulting in bounded constraint violations. Based on these endeavors, we propose a novel and general graph attention mechanism (25; 10): Graph Sinkhorn Attention (GSINA) for improving GIL tasks of multiple levels, GSINA evaluates the graph features (nodes and edges) importance by assigning sparse and soft graph attention values. As an invariance optimization method, GSINA defines its invariant subgraph in the manner of graph attention, which serves as a powerful regularization to improve GIL.

Therefore, our contributions are as follows successively:

**1]** To the best of our knowledge, we are the first to point out the necessity of sparsity, softness, and differentiability in subgraph extracting for GIL, lacking in previous IB and top-$k$ based methods.

**2]** We propose Graph Sinkhorn Attention (GSINA), a GIL framework by learning fully differentiable invariant subgraphs with controllable sparsity and softness to improve multiple levels of graph generalization tasks.

**3]** Extensive experiments validate the superiority of our GSINA, which could outperform the state-of-the-art GIL methods GSAT (10), CIGA (11), and EERM (15) by large margins.

The related works cover different aspects of our work, including the problem formulation of graph out-of-distribution (OOD) generalization and Graph Invariant Learning (GIL), the inductive bias behind invariant subgraph extraction, and the cardinality-constrained combinatorial optimization for a fully differentiable top-$k$ operation, which we leave to Appendix A.1 due to page limit.

## 2  APPROACH

In this section, we will first introduce the learning objective of our Graph Sinkhorn Attention (GSINA) for Graph Invariant Learning (GIL) at a high level, then the implementation details of GSINA: the utilization of the Sinkhorn algorithm to obtain the sparse, soft, and differentiable invariant subgraph as a kind of graph attention mechanism, and the general representation learning framework for GIL of multiple level tasks. More detailedly derivations and implementation of GSINA can be found in Appendix A and C.

### 2.1  GRAPH SINKHORN ATTENTION: LEARNING OBJECTIVE

Aiming at finding the invariant subgraph $G_S$ with a stable relationship to the label $Y$, we formulate it as a mutual information $I(G_S; Y)$ maximization problem like the practices in (20; 10; 11). On the other hand, constraints should be applied on the invariant subgraph $G_S$ to ensure its informative conciseness (i.e., the information of the variant or redundant part of the input graph $G$ should be damped), the constraints act as regularizations and improve the generalization of graph learning tasks. As we discussed in Sec. 1, the information bottleneck (IB) based methods (20; 10)might hardly guarantee the subgraph conciseness (lack of sparsity), and the partially differentiable top-$k$ selection based methods (9; 11) generate hard subgraphs and shrink the subgraph solution space. Differently, we leverage a subgraph extractor $g_\phi(G, r, \tau)$ generating softly cardinality-constrained subgraph $G_S$, the subgraph ratio $r$ is the cardinality constraint controlling the sparsity of $G_S$, and $\tau$ is a temperature hyperparameter controlling the softness of $G_S$, which will be detailedly discussed in Sec. 2.2.

Our subgraph extractor $g_\phi(G, r, \tau)$ acts as a sparsity and softness regularization of $G_S$, and the mutual information maximization problem can be formulated as:

$$\max_{\phi} I\left(G_S; Y\right), \text{ s.t. } G_S \sim g_\phi(G, r, \tau). \tag{1}$$

As the direct estimation of mutual information $I(G_S; Y)$ is intractable, we derive its lower bound with the help of a variational approximation distribution $P_\theta(Y|G_S)$ parameterized by $\theta$, which also acts as the predictor of label $Y$ given the invariant subgraph $G_S$:

$$I\left(G_S; Y\right) \geq \mathbb{E}_{G_S, Y}\left[\log P_\theta(Y|G_S)\right] + H(Y), \tag{2}$$

where $H(Y)$ is a constant entropy of the label distribution $P(Y)$ and can be omitted in the optimization, the problem in Eq. 1 can be optimized by maximizing the lower bound item in Eq. 2 and the final learning objective of GSINA is:

$$\max_{\theta, \phi} \mathbb{E}_{G_S, Y}\left[\log P_\theta(Y|G_S)\right], \text{ s.t. } G_S \sim g_\phi(G, r, \tau). \tag{3}$$

It results in a two-stage forward pipeline: first extracting the invariant subgraph $G_S$ from the input graph $G$, then making prediction $Y$ based on $G_S$. Although sharing similarities, GSINA is unlike the IB based methods (20; 10): the subgraph information of GSINA is constrained by our subgraph extractor $g_\phi(G, r, \tau)$, which explicitly controls the sparsity $r$ and softness $\tau$ of the invariant subgraph.

## 2.2 GRAPH SINKHORN ATTENTION: IMPLEMENTATION

To extract a sparse, soft, and differentiable invariant subgraph $G_S$ from the input graph $G$, we leverage a softly cardinality-constrained subgraph extractor $g_\phi(G, r, \tau)$ based on the Sinkhorn Algorithm.

The goal of graph attention mechanisms (25; 10) is to assign attention coefficients to different parts of the input graph structure, evaluating their respective importance for the prediction of the target label $Y$. Beyond the graph attention mechanisms designed in (25; 10), we take the sparsity and softness of the attention distribution into consideration by applying differentiable top-$k$ (26; 22) to evaluate the importance of edges. According to Sec. A.1.2, it has a corresponding popular OT-theoretic solution of the Sinkhorn algorithm (24), and the Gumbel re-parameterization trick could be adopted to enhance the performance according to the practice in (22). For edge attention, GSINA softly highlights the top-$r$ ratio most influential edges and 'filters out' other edges by assigning sparse edge attention to the input graph $G$ (as shown in Fig. 1, 2) and provides soft attention distribution. Based on GSINA edge attention, sparse and soft GSINA node attention could be designed based on graph neighborhood aggregation to evaluate the importance of nodes.

We will start by describing the implementation of edge and node attention in GSINA, and then the general framework for multiple-level GIL tasks via GSINA.

**Edge Attention.** As an initial step, a composition of $\text{GNN}_\phi$ and $\text{MLP}_\phi$ is leveraged to obtain learnable node features $\{\mathbf{h}_i | i \in \mathcal{V}\}$ and edge scores $s = \{s_e | e \in \mathcal{E}\}$ of the input graph $G = (\mathcal{V}, \mathcal{E})$:

$$\{\mathbf{h}_i\} = \text{GNN}_\phi(G), \ i \in \mathcal{V}, \quad s_e = \text{MLP}_\phi(\mathbf{h}_i, \mathbf{h}_j), \ e = (i, j). \tag{4}$$

Softly selecting the top-$r$ scored edges as the invariant subgraph could be interpreted as a relaxed OT problem, whose setting is to move $rN_e$ items to the destination of the invariant part, and the other $(1-r)N_e$ elements to the other destination of the variant part, where $N_e$ is the number of the edges. During the training phase, the Gumbel re-parameterization trick $\tilde{s}_e = s_e - \sigma \log(-\log u_e), u_e \sim U(0, 1)$ (27; 22), where $\sigma$ is the factor of Gumbel noise and $\sigma = 0$ in validation and testing phases, could be adopted to enlarge the sampling space and to improve the generalization performances, i.e. the Gumbel trick allows less important edges to participate in training (without being poorly trained due to low attention, resulting in underfitting), and remains the sampling accuracy. Defining $\mathbf{D}$ as the distance matrix of the OT problem, $\mathbf{R}$ and $\mathbf{C}$ as the marginal distributions, and $\mathbf{T}$ as the transportation plan moving $rN_e$ items to $\max(s)$ (invariant) and $(1-r)N_e$ items to $\min(s)$ (variant), the OT problem for GSINA edge attention can be formulated as follows:

$$\begin{aligned} \mathbf{D} &= \begin{bmatrix} \tilde{s}_1 - \min(s), & \tilde{s}_2 - \min(s), & \dots, & \tilde{s}_{N_e} - \min(s) \\ \max(s) - \tilde{s}_1, & \max(s) - \tilde{s}_2, & \dots, & \max(s) - \tilde{s}_{N_e} \end{bmatrix}, \\ \mathbf{R} &= [(1-r)N_e, \ rN_e]^\top, \quad \mathbf{C} = [1, 1, \dots, 1]^\top \in \mathbb{R}^{N_e \times 1}, \\ \min_{\mathbf{T}} &\operatorname{tr}(\mathbf{T}^\top \mathbf{D})\mathbf{e} - \tau H(\mathbf{T}), \ \text{s.t.} \ \mathbf{T} \in [0, 1]^{2 \times N_e}, \ \mathbf{T}\mathbf{1} = \mathbf{R}, \ \mathbf{T}^\top \mathbf{1} = \mathbf{C}, \end{aligned} \tag{5}$$

which is relaxed by the entropic regularizer (28) of $\tau H(\mathbf{T})$, the softness of the transportation plan $\mathbf{T}$ could be regularized by setting the temperature hyperparameter $\tau$. The transportation plan $\mathbf{T}$ could be iteratively solved by the Sinkhorn algorithm:

$$\mathbf{T}_0 = \exp\left(-\frac{\mathbf{D}}{\tau}\right), \ \mathbf{T}_k = \operatorname{diag}\left(\mathbf{T}_{k-1}\mathbf{1} \oslash \mathbf{R}\right)^{-1} \mathbf{T}_{k-1}, \ \mathbf{T}_k = \mathbf{T}_{k-1} \operatorname{diag}(\mathbf{T}_{k-1}^\top \mathbf{1} \oslash \mathbf{C})^{-1}, \tag{6}$$

where $\mathbf{T}_0$ is the initialization, the equations of $\mathbf{T}_k$ are alternative iterations of row- and column-wise normalizations to satisfy the two constraints $\mathbf{T}\mathbf{1} = \mathbf{R}$ and $\mathbf{T}^\top \mathbf{1} = \mathbf{C}$, $\oslash$ is element-wise division.

Eventually, the edge attention $\alpha^E = \{\alpha_e^E | e \in \mathcal{E}\}$ could be obtained from the procedure above:

$$\left[\alpha_1^E, \alpha_2^E, \dots, \alpha_{N_e}^E\right] = \mathbf{T}[1, :]. \tag{7}$$

**Node Attention.** Given the sparse, soft, and differentiable edge attention $\alpha^E$, a natural consideration is to evaluate the importance of nodes. Hence, the node attention $\alpha^V = \{\alpha_i^V | i \in \mathcal{V}\}$ in our GSINA is proposed, which could be obtained by an aggregation of the edge attention in the neighborhood of each node $i$:

$$\alpha_i^V = \text{AGG}(\{\alpha_e^E | e = (i, j) \wedge e \in \mathcal{E}\}). \tag{8}$$

For the 'hard' top-$k$ based GIL methods (9; 11), only the selected part is kept to be the invariant subgraph $G_S$ and the other part $\overline{G_S}$ is just discarded. In other words, the node importance is 1 for nodes in $G_S$ and 0 for nodes in $\overline{G_S}$. Our node attention is a soft and fully differentiable version to mimic their invariant subgraph extractions.

**General Graph Invariant Learning.** Our invariant subgraph extraction results in a graph weighted by the Graph Sinkhorn (Edge and Node) Attention, with the properties of sparsity, softness, and differentiability, the mathematical definition of the invariant subgraph $G_S$ of our GSINA is in the manner of graph attention:

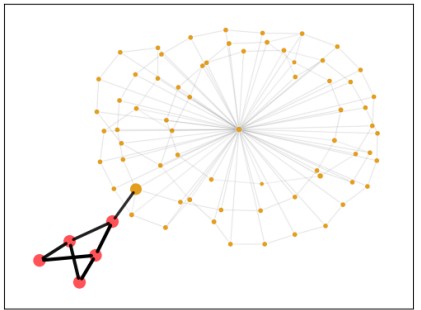

$$G_S = \{G, \alpha^V, \alpha^E\} \sim g_\phi(G, r, \tau) \qquad (9)$$

Based on the definition of our invariant subgraph in Eq. 9, the prediction process (the predictor $P_\theta(Y|G_S)$ in Eq. 3) could be regularized by our GSINA message passing mechanism in Eq. 10. For the $l$-th GNN message passing layer, GSINA weights each message $m_\theta(\mathbf{h}_i^{(l)}, \mathbf{h}_j^{(l)}, \mathbf{h}_{ij}^{(l)})$ by edge attention $\alpha_{ij}^E$, $\mathbf{h}_{ij}^{(l)}$ is the representation of edge $(i, j)$ (if applicable), $\bigoplus$ is any permutation invariant aggregation function, $\gamma_\theta$ is the GNN update function, and $\mathbf{h}_i^{(l+1)}$ is updated representation. If the graph representation $\mathbf{h}_G$ is obtained from a readout $f_\theta$ of node representations output from the $L$-th (final) GNN layer $\{\mathbf{h}_i^{(L)}|i \in \mathcal{V}\}$, each node representation $\mathbf{h}_i^{(L)}$ is weighted by our node attention $\alpha_i^V$ in our GSINA, GSINA is general due to its applicability to multiple-level (i.e. graph-level and node-level) GIL tasks:

Figure 2: Example of the invariant subgraph $G_S$ weighted by GSINA from SPMotif dataset. The ground truth of the invariant subgraph is colored red, and the other part is yellow. The edge widths and node sizes are given by GSINA original outputs (we do not apply attention scaling tricks for visualizations like GSAT (10) and CIGA (29)). It is shown our GSINA assigns sparse and soft attention $\{\alpha^V, \alpha^E\}$ to the nodes and edges of the input graph $G$. Interpretability analysis and more visualization results can be found in Appendix D.

$$\mathbf{h}_i^{(l+1)} = \gamma_\theta \left( \mathbf{h}_i^{(l)}, \bigoplus_{j \in \mathcal{N}_i} \alpha_{ij}^E * m_\theta(\mathbf{h}_i^{(l)}, \mathbf{h}_j^{(l)}, \mathbf{h}_{ij}^{(l)}) \right), \quad \mathbf{h}_G = f_\theta \left( \{\alpha_i^V * \mathbf{h}_i^{(L)}|i \in \mathcal{V}\} \right). \quad (10)$$

## 3 EXPERIMENTS

Experiments are conducted on various benchmarks following GSAT (10), CIGA (11), and EERM (15) to evaluate GSINA's effectiveness for different graph learning tasks, including graph-level OOD generalization and node-level. In this section, we introduce the datasets, baselines, evaluation metrics, and experiment settings and provide results analysis. More experiment settings can be found in Appendix B, and more analysis including model selection and interpretation performance analysis can be found in Appendix D. All experiments are conducted for 5 runs on RTX-2080Ti (11GB) GPUs, and the average and standard deviation are reported.

### 3.1 ON GRAPH-LEVEL GRAPH INVARIANT LEARNING TASKS

For direct and fair comparisons with the two GIL SOTAs, GSAT (10) and CIGA (11), each respectively using different datasets and GNN backbones, we perform evaluations strictly in line with their original corresponding settings and the results are given in Tab. 2, 3 and Tab. 4, 5 respectively. To compare with GSAT, the hyperparameter $r$ for GSINA is chosen according to the validation performances. For CIGA, as CIGA also performs top-$r$ subgraph extractions, $r$ follows the settings of CIGA on the datasets in Tab. 4, 5.

**Datasets.** To compare with GSAT, we use the synthetic Spurious-Motif (SPMotif) datasets from DIR (9), where each graph is constructed by a combination of a motif graph directly determining the graph label, and a base graph providing spurious correlation to graph label, and we use datasets with spurious correlation degree $b = 0.5, 0.7$ and $0.9$. For real-world datasets, we use MNIST-75sp (30), where each image in MNIST is converted to a superpixel graph, Graph-SST2 (31; 32), which is a sentiment analysis dataset, and each text sequence in SST2 is converted to a graph, following the

Table 2: Graph-level OOD generalization performances (GSAT (10) benchmark).

| | MOLHIV (AUC) | GRAPH-SST2 | MNIST-75SP | SPURIOUS-MOTIF | | |
| | | | | $b = 0.5$ | $b = 0.7$ | $b = 0.9$ |
|---|---|---|---|---|---|---|
| GIB (20) | $76.43_{\pm2.65}$ | $82.99_{\pm0.67}$ | $93.10_{\pm1.32}$ | $54.36_{\pm7.09}$ | $48.51_{\pm5.76}$ | $46.19_{\pm5.63}$ |
| DIR (9) | $76.34_{\pm1.01}$ | $82.32_{\pm0.85}$ | $88.51_{\pm2.57}$ | $45.49_{\pm3.81}$ | $41.13_{\pm2.62}$ | $37.61_{\pm2.02}$ |
| GIN (6) | $76.69_{\pm1.25}$ | $82.73_{\pm0.77}$ | $95.74_{\pm0.36}$ | $39.87_{\pm1.30}$ | $39.04_{\pm1.62}$ | $38.57_{\pm2.31}$ |
| GIN+GSAT (10) | $76.47_{\pm1.53}$ | $82.95_{\pm0.58}$ | $96.24_{\pm0.17}$ | $52.74_{\pm4.08}$ | $49.12_{\pm3.29}$ | $44.22_{\pm5.57}$ |
| GIN+OURS | $\mathbf{77.99}_{\pm0.97}$ | $\mathbf{83.66}_{\pm0.37}$ | $\mathbf{96.73}_{\pm0.16}$ | $\mathbf{55.16}_{\pm5.69}$ | $\mathbf{56.83}_{\pm6.32}$ | $\mathbf{49.86}_{\pm6.10}$ |
| PNA (39) | $78.91_{\pm1.04}$ | $79.87_{\pm1.02}$ | $87.20_{\pm5.61}$ | $68.15_{\pm2.39}$ | $66.35_{\pm3.34}$ | $61.40_{\pm3.56}$ |
| PNA+GSAT (10) | $80.24_{\pm0.73}$ | $80.92_{\pm0.66}$ | $93.96_{\pm0.92}$ | $68.74_{\pm2.24}$ | $64.38_{\pm3.20}$ | $57.01_{\pm2.95}$ |
| PNA+OURS | $\mathbf{80.55}_{\pm0.97}$ | $\mathbf{82.18}_{\pm1.01}$ | $\mathbf{95.48}_{\pm0.37}$ | $\mathbf{76.39}_{\pm1.85}$ | $\mathbf{73.96}_{\pm2.87}$ | $\mathbf{62.51}_{\pm5.86}$ |

Table 3: Graph-level OOD generalization performances (other OGBG-Mol datasets in GSAT (10) benchmark).

| | MOLBACE | MOLBBBP | MOLCLINTOX | MOLTOX21 | MOLSIDER |
|---|---|---|---|---|---|
| PNA (39) | $73.52_{\pm3.02}$ | $67.21_{\pm1.34}$ | $86.72_{\pm2.33}$ | $75.08_{\pm0.64}$ | $56.51_{\pm1.90}$ |
| GSAT (10) | $77.41_{\pm2.42}$ | $\mathbf{69.17}_{\pm1.12}$ | $87.80_{\pm2.36}$ | $74.96_{\pm0.66}$ | $57.58_{\pm1.23}$ |
| OURS | $\mathbf{79.57}_{\pm1.38}$ | $67.86_{\pm0.91}$ | $\mathbf{90.08}_{\pm2.06}$ | $\mathbf{75.47}_{\pm0.55}$ | $\mathbf{58.61}_{\pm1.09}$ |

splits in DIR (9), Graph-SST2 contains degree shifts, and molecular property prediction datasets from the OGBG (33; 34) benchmark (molhiv, molbace, molbbbp, molclintox, moltox21, molsider).

To compare with CIGA (11), we also use the synthetic SPMotif datasets from DIR, with structural shift degrees $b = 0.33$, 0.6 and 0.9, denoted as SPMotif (-struc). Besides, we use the SPMotif (-mixed) from CIGA (11), whose distribution shifts are additionally mixed with attribute shifts. For real-world datasets, in line with CIGA, to validate the generalization performance with more complicated relationships under distribution shifts, we use sentiment analysis datasets Graph-SST5 (32) and Twitter (35) with degree shifts, DrugOOD datasets (36), which is from AI-aided Drug Discovery, the split schemes including assay, scaffold and size, and the datasets from TU (37) benchmarks (nci1, nci109, proteins, dd) to examine the OOD generalization under graph size shifts.

**Metrics.** We test the classification accuracy (ACC) for SPMotif datasets, MNIST-75sp, Graph-SST2, Graph-SST5, Twitter, ROC-AUC for OGBG and DrugOOD datasets, and Matthews correlation coefficient (MCC) for TU datasets following (38; 10; 11).

**Baselines.** Following GSAT (10)'s settings, we compare with interpretable GNNs GIB (20) and DIR (9), and we use GIN (6) and PNA (39) as backbones for GSAT and GSINA. Following CIGA (11), in addition to ERM (40), we also compare with interpretable GNNs GIB, DIR, ASAP Pooling (41), as well as the invariant learning methods IRM (42), V-Rex (43), IB-IRM (44), EIIL (45) and CNC (46), and the Oracle (IID) performances on the datasets without distribution shifts are also reported, we use the GNN architectures in line with CIGA to test GSINA.

**Performances Analysis.** Tab. 2 reports the graph classification performances on GSAT benchmark, GSINA achieves better performances than the baselines of interpretable GNNs GIB, DIR, and GSAT. GSINA outperforms GSAT by large margins on all 6 datasets for both GNN backbones GIN and PNA. Tab. 3 reports the graph classification performances for another 5 OGBG-Mol datasets with smaller sizes than those in Tab. 2. Following GSAT, we compare with PNA backboned baselines, and GSINA mostly outperforms.

Tab. 4, 5 report the OOD generalization performances on CIGA benchmark. Our GSINA outperforms all the baselines of interpretable GNNs (ASAP, GIB, and DIR) by large margins. For the invariant learning baselines, our GSINA also achieve better performances. Comparing with CIGA, GSINA achieves the best performances on SPMotif (except for -struct, $b = 0.33$), Graph-SST5, Twitter, proteins and dd. On nci1, nci109, and DrugOOD datasets, GSINA produces results comparable to CIGA, indicating the difficulties of these OOD generalization tasks, meanwhile, the improvements of CIGA compared to ERM on DrugOOD datasets are also by little margins.

These improvements demonstrate the effectiveness of our sparsity compared with GSAT (based on IB constraint) and softness compared with CIGA (based on top-$k$). Especially, we observe relatively big improvements on SPMotif datasets. Respectively, there are about 10% and 15% improvements than GSAT and CIGA in classification accuracy on SPMotif ($b = 0.7$) in Tab. 2 and SPMotif (-mixed, $b = 0.9$) in Tab. 4, which indicates the superiority of our GSINA on the datasets with more

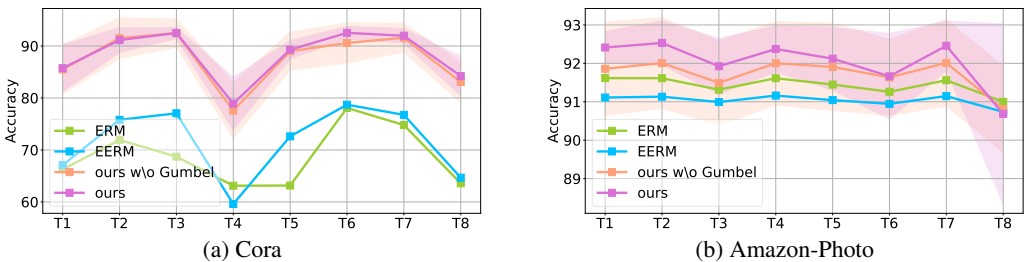

Figure 3: Node-level OOD generalization performances for 'Artificial Transformation'.

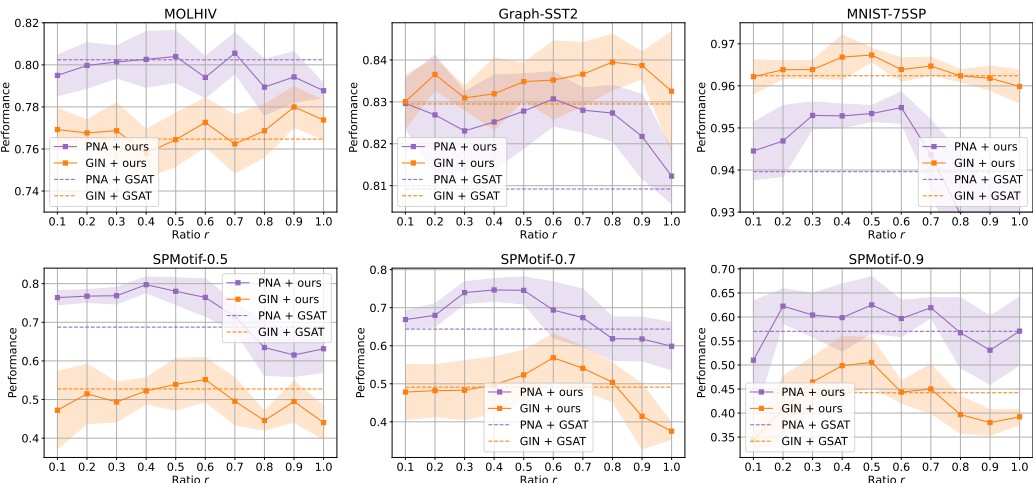

Figure 4: Graph-level OOD generalization performances for different hyperparameter $r$ in GSINA.

Table 4: Graph-level OOD generalization performances (synthetic datasets in CIGA benchmark).

|  | SPMotif-Struc | | | SPMotif-Mixed | | |
|---|---|---|---|---|---|---|
|  | BIAS=0.33 | BIAS=0.60 | BIAS=0.90 | BIAS=0.33 | BIAS=0.60 | BIAS=0.90 |
| ERM (40) | $59.49_{\pm3.50}$ | $55.48_{\pm4.84}$ | $49.64_{\pm4.63}$ | $58.18_{\pm4.30}$ | $49.29_{\pm8.17}$ | $41.36_{\pm3.29}$ |
| IRM (42) | $57.15_{\pm3.98}$ | $61.74_{\pm1.32}$ | $45.68_{\pm4.88}$ | $58.20_{\pm1.97}$ | $49.29_{\pm3.67}$ | $40.73_{\pm1.93}$ |
| V-REX (43) | $54.64_{\pm3.05}$ | $53.60_{\pm3.74}$ | $48.86_{\pm9.69}$ | $57.82_{\pm5.93}$ | $48.25_{\pm2.79}$ | $43.27_{\pm1.32}$ |
| EIIL (45) | $56.48_{\pm2.56}$ | $60.07_{\pm4.47}$ | $55.79_{\pm6.54}$ | $53.91_{\pm3.15}$ | $48.41_{\pm5.53}$ | $41.75_{\pm4.97}$ |
| IB-IRM (44) | $58.30_{\pm6.37}$ | $54.37_{\pm7.35}$ | $45.14_{\pm4.07}$ | $57.70_{\pm2.11}$ | $50.83_{\pm1.51}$ | $40.27_{\pm3.68}$ |
| CNC (46) | $70.44_{\pm2.55}$ | $66.79_{\pm9.42}$ | $50.25_{\pm10.7}$ | $65.75_{\pm4.35}$ | $59.27_{\pm5.29}$ | $41.58_{\pm1.90}$ |
| ASAP (41) | $64.87_{\pm13.8}$ | $64.85_{\pm10.6}$ | $57.29_{\pm14.5}$ | $66.88_{\pm15.0}$ | $59.78_{\pm6.78}$ | $50.45_{\pm4.90}$ |
| DIR (9) | $58.73_{\pm11.9}$ | $48.72_{\pm14.8}$ | $41.90_{\pm9.39}$ | $67.28_{\pm4.06}$ | $51.66_{\pm14.1}$ | $38.58_{\pm5.88}$ |
| CIGAv1 (11) | $71.07_{\pm3.60}$ | $63.23_{\pm9.61}$ | $51.78_{\pm7.29}$ | $74.35_{\pm1.85}$ | $64.54_{\pm8.19}$ | $49.01_{\pm9.92}$ |
| CIGAv2 (11) | $\mathbf{77.33}_{\pm9.13}$ | $69.29_{\pm3.06}$ | $63.41_{\pm7.38}$ | $72.42_{\pm4.80}$ | $70.83_{\pm7.54}$ | $54.25_{\pm5.38}$ |
| OURS | $75.49_{\pm4.26}$ | $\mathbf{74.25}_{\pm2.53}$ | $\mathbf{73.54}_{\pm5.54}$ | $\mathbf{82.70}_{\pm6.28}$ | $\mathbf{77.03}_{\pm2.66}$ | $\mathbf{68.89}_{\pm8.17}$ |
| ORACLE (IID) | $88.70_{\pm0.17}$ | $88.70_{\pm0.17}$ | $88.70_{\pm0.17}$ | $88.73_{\pm0.25}$ | $88.73_{\pm0.25}$ | $88.73_{\pm0.25}$ |

distinguishable subgraph properties, such as the SPMotif datasets, where the invariant subgraphs can be clearly separable due to their data generation processes.

**Hyperparameter Studies.** As shown in Fig. 4, for the 6 datasets used in the GSAT benchmark and reported in Tab. 2, there are rough 'increasing-decreasing' patterns in the curve of predictive performance and subgraph ratio (i.e. sparsity) $r$, which reflects our GSINA is sensible to the hyperparameter $r$. The patterns of the curves in Fig. 4 show that a too-small or a too-large $r$ results in worse generalization. When $r$ is too small, it is more likely that the extracted subgraph $G_S$ is too sparse and lacks information; when $r$ is too large, it also results in a $G_S$ lack of information, as more redundant parts of the input graph $G$ would be selected, which results in a $G_S$ with little difference with $G$, and when $r = 1.0$, GSINA degenerates to ERM with all edge attention set to 1. The hyperparameters of Sinkhorn temperature $\tau$ and Gumbel noise factor $\sigma$ are not-tuned and both are set to 1 in all experiments, we show the reasonability of their settings in Appendix D.

Table 5: Graph-level OOD generalization performances (real-world datasets in CIGA (11) benchmark), **bold** font for the best performance on each dataset and underline for the second.

| | GRAPH-SST5 | TWITTER | DRUG-ASSAY | DRUG-SCA | DRUG-SIZE | NCI1 | NCI109 | PROT | DD |
|---|---|---|---|---|---|---|---|---|---|
| ERM (40) | $43.89_{\pm1.73}$ | $60.81_{\pm2.05}$ | $71.79_{\pm0.27}$ | $68.85_{\pm0.62}$ | $66.70_{\pm1.08}$ | $0.15_{\pm0.05}$ | $0.16_{\pm0.02}$ | $0.22_{\pm0.09}$ | $0.27_{\pm0.09}$ |
| IRM (42) | $43.69_{\pm1.26}$ | $63.50_{\pm1.23}$ | $72.12_{\pm0.49}$ | $68.69_{\pm0.65}$ | $66.54_{\pm0.42}$ | $0.17_{\pm0.02}$ | $0.14_{\pm0.01}$ | $0.21_{\pm0.09}$ | $0.22_{\pm0.08}$ |
| V-REX (43) | $43.28_{\pm0.52}$ | $63.21_{\pm1.57}$ | $72.05_{\pm1.25}$ | $68.92_{\pm0.98}$ | $66.33_{\pm0.74}$ | $0.15_{\pm0.04}$ | $0.15_{\pm0.04}$ | $0.22_{\pm0.06}$ | $0.21_{\pm0.07}$ |
| EIIL (45) | $42.98_{\pm1.03}$ | $62.76_{\pm1.72}$ | $72.60_{\pm0.47}$ | $68.45_{\pm0.53}$ | $66.38_{\pm0.66}$ | $0.14_{\pm0.03}$ | $0.16_{\pm0.02}$ | $0.20_{\pm0.05}$ | $0.23_{\pm0.10}$ |
| IB-IRM (44) | $40.85_{\pm2.08}$ | $61.26_{\pm1.20}$ | $72.50_{\pm0.49}$ | $68.50_{\pm0.40}$ | $66.64_{\pm0.28}$ | $0.12_{\pm0.04}$ | $0.15_{\pm0.06}$ | $0.21_{\pm0.06}$ | $0.15_{\pm0.13}$ |
| CNC (46) | $42.78_{\pm1.53}$ | $61.03_{\pm2.49}$ | $72.40_{\pm0.46}$ | $67.24_{\pm0.90}$ | $65.79_{\pm0.80}$ | $0.16_{\pm0.04}$ | $0.16_{\pm0.04}$ | $0.19_{\pm0.08}$ | $0.27_{\pm0.13}$ |
| ASAP (41) | $44.16_{\pm1.36}$ | $60.68_{\pm2.10}$ | $70.51_{\pm1.93}$ | $66.19_{\pm0.94}$ | $64.12_{\pm0.67}$ | $0.16_{\pm0.10}$ | $0.15_{\pm0.07}$ | $0.22_{\pm0.16}$ | $0.21_{\pm0.08}$ |
| GIB (20) | $38.64_{\pm4.52}$ | $48.08_{\pm2.27}$ | $63.01_{\pm1.16}$ | $62.01_{\pm1.41}$ | $55.50_{\pm1.42}$ | $0.13_{\pm0.10}$ | $0.16_{\pm0.02}$ | $0.19_{\pm0.08}$ | $0.01_{\pm0.18}$ |
| DIR (9) | $41.12_{\pm1.96}$ | $59.85_{\pm2.98}$ | $68.25_{\pm1.40}$ | $63.91_{\pm1.36}$ | $60.40_{\pm1.42}$ | $0.21_{\pm0.06}$ | $0.13_{\pm0.05}$ | $0.25_{\pm0.14}$ | $0.20_{\pm0.10}$ |
| CIGAv1 (11) | $44.71_{\pm1.14}$ | $63.66_{\pm0.84}$ | $72.71_{\pm0.52}$ | $69.04_{\pm0.86}$ | $67.24_{\pm0.88}$ | $0.22_{\pm0.07}$ | $\mathbf{0.23}_{\pm0.09}$ | $\underline{0.40}_{\pm0.06}$ | $\mathbf{0.29}_{\pm0.08}$ |
| CIGAv2 (11) | $\underline{45.25}_{\pm1.27}$ | $\underline{64.45}_{\pm1.99}$ | $\mathbf{73.17}_{\pm0.39}$ | $\mathbf{69.70}_{\pm0.27}$ | $\mathbf{67.78}_{\pm0.76}$ | $\underline{0.27}_{\pm0.07}$ | $\underline{0.22}_{\pm0.05}$ | $0.31_{\pm0.12}$ | $0.26_{\pm0.08}$ |
| OURS | $\mathbf{45.84}_{\pm0.52}$ | $\mathbf{64.64}_{\pm1.71}$ | $\underline{72.84}_{\pm0.50}$ | $\underline{69.57}_{\pm0.39}$ | $67.48_{\pm0.33}$ | $\mathbf{0.28}_{\pm0.07}$ | $0.21_{\pm0.04}$ | $\mathbf{0.41}_{\pm0.07}$ | $\mathbf{0.29}_{\pm0.07}$ |
| ORACLE (IID) | $48.18_{\pm1.00}$ | $64.21_{\pm1.77}$ | $85.56_{\pm1.44}$ | $84.71_{\pm1.60}$ | $85.83_{\pm1.31}$ | $0.32_{\pm0.05}$ | $0.37_{\pm0.06}$ | $0.39_{\pm0.09}$ | $0.33_{\pm0.05}$ |

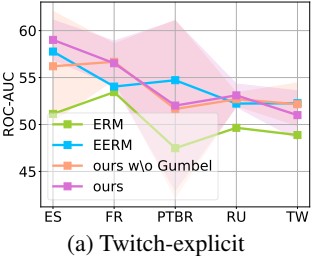

(a) Twitch-explicit

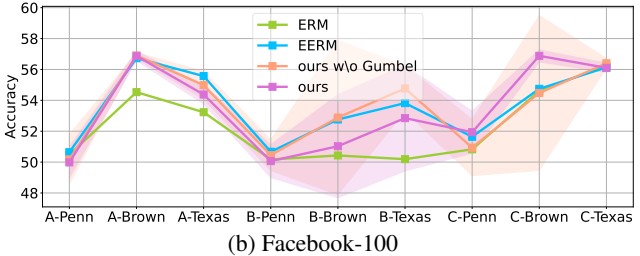

(b) Facebook-100

Figure 5: Node-level OOD generalization performances for 'Cross-Domain Transfers'.

## 3.2 ON NODE-LEVEL GRAPH INVARIANT LEARNING TASKS

**Datasets.** We follow the datasets and protocols used in EERM (15), which involve 3 types of distribution shifts: 1) "Artificial Transformation": synthetic spurious features are added to Cora and Amazon-Photo, and there are 8 testing graphs (T1 ∼ T8) for both datasets, 2) "Cross-Domain Transfers": each graph in Twitch-explicit and Facebook-100 corresponds to distinct domains; for Twitch-explicit, DE is used for training, ENGB for validation and 5 graphs ES, FR, PTBR, RU, TW for testing; for Facebook-100, 3 different training sets are used, we denote them as A = (Johns Hopkins, Caltech, Amherst), B = (Bingham, Duke, Princeton), C = (WashU, Brandeis, Carnegie), the validation set is (Cornell, Yale), and the testing set is (Penn, Brown, Texas), 3) "Temporal Evolution": train/val/test splits for Elliptic and OGB-Arxiv are made by time, Elliptic provides 9 test graphs (T1 ∼ T9), and OGB-Arxiv provides 3 time windows (14-16, 16-18, 18-20).

**Baselines.** We compare our GSINA with ERM (40) and EERM (15), following the settings in EERM, we use GCN (47) as backbone subgraph extractors, predictors and spurious features generators (if applicable) for Cora, Amazon-Photo, Twitch-explicit and Facebook-100, SAGE (48) for Elliptic and OGB-Arxiv. According to the hyperparameter studies in Sec. 3.1, we regard $r = 0.5$ as a reasonable choice and set it for all node classification experiments with our GSINA.

**Metrics.** We test the node classification accuracy (ACC) on Cora, Amazon-Photo, Facebook-100, OGB-Arxiv, ROC-AUC for Twitch-explicit, and F1-score for Elliptic.

**Performances Analysis.** Fig. 3, 5, 6 reports the generalization performance for the distribution shifts of 'Artificial Transformation', 'Cross-Domain Transfers', and 'Temporal Evolution', respectively. Under most testing scenarios, our GSINA outperforms ERM for node classification. We achieve better results than EERM on Cora, Amazon-Photo, Elliptic, and comparable results to EERM on Twitch-explicit, Facebook-100, and OGB-Arxiv. Especially, our GSINA achieves an improvement of about 20% for classification accuracy on Cora.

## 3.3 ABLATION STUDIES

Here we provide our ablation studies on both graph and node classification tasks. Tab. 6 reports detailed experiment results on SPMotif datasets in GSAT benchmark, the ablation versions of our GSINA are without (w/o) the Gumbel noise, Node Attention, or both (denoted as G & N), while the hyperparameter $r$ remain unchanged. As GSINA does not consider the Node Attention for node classification tasks, the ablation studies in Fig. 3, 5, 6 only provide the version without the Gumbel noise.

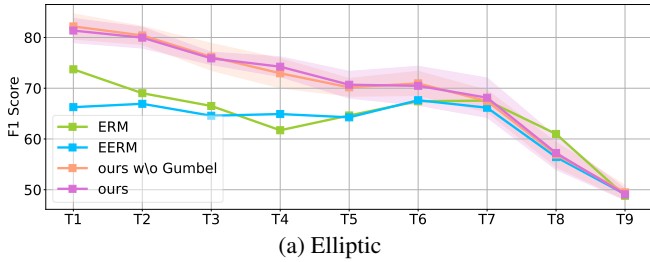
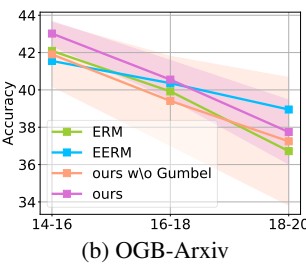

(a) Elliptic  (b) OGB-Arxiv

Figure 6: Node-level OOD generalization performances for Temporal Evolution datasets.

From the ablation studies, we observe performance degradations for graph and node level tasks when learning GSINA without Gumbel noise, Node Attention, or both, demonstrating the effectiveness of the proposed components of GSINA. Moreover, in the node classification tasks, the ablation versions provide higher variances on Cora, Amazon-Photo, and OGB-Arxiv, which demonstrate the Gumbel trick could stabilize the learning of GSINA.

Table 6: Ablation studies on SPMotif datasets.

| | SPURIOUS-MOTIF | | |
| --- | --- | --- | --- |
| | $b = 0.5$ | $b = 0.7$ | $b = 0.9$ |
| GIN+OURS | **55.16**±5.69 | **56.83**±6.32 | 49.86±6.10 |
| W/O GUMBEL | 48.27±4.80 | 45.25±7.15 | **50.28**±2.83 |
| W/O NODEATTN | 47.34±7.99 | 54.63±6.99 | 48.41±1.16 |
| W/O G&N | 46.28±5.67 | 45.40±3.22 | 44.44±6.56 |
| GIN+GSAT | 52.74±4.08 | 49.12±3.29 | 44.22±5.57 |
| PNA+OURS | **76.39**±1.85 | **73.96**±2.87 | **62.51**±5.86 |
| W/O GUMBEL | 69.95±2.76 | 69.67±3.44 | 62.14±4.64 |
| W/O NODEATTN | 71.60±1.89 | 58.70±3.82 | 58.20±2.70 |
| W/O G&N | 71.75±2.42 | 67.50±4.51 | 61.34±1.72 |
| PNA+GSAT | 68.74±2.24 | 64.38±3.20 | 57.01±2.95 |

## 4 CONCLUSION

In this paper, we have proposed Graph Sinkhorn Attention (GSINA), a general invariance optimization framework for Graph Invariant Learning (GIL) to improve the generalization for both graph and node level tasks by extracting the sparse, soft, and differentiable invariant subgraphs in the manner of graph attention. Extensive experiments have shown the superiority of GSINA against the state-of-the-arts on both graph and node level GIL tasks.

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

APPENDIX

# A THEORETIC DETAILS

## A.1 BACKGROUND AND RELATED WORKS

### A.1.1 GRAPH INVARIANT LEARNING

The concept of invariant learning involves utilizing the invariant relationships between features and labels across various distributions while disregarding any spurious correlations that may arise (7). Through this approach, it is possible to attain a high level of out-of-distribution (OOD) generalization in the presence of distribution shifts.

Graph OOD problem considers a series of graph datasets $\mathcal{G} = \{\mathcal{G}^e\}_e$ collected from multiple environments $\mathcal{E}_{\text{all}}$. For each dataset $\mathcal{G}^e = \{(G_i^e, Y_i^e)\}_{i=1}^{N^e}$ collected from environment $e$, $(G_i^e, Y_i^e)$ is a graph-label pair sampled from it and $N^e$ is the number of such pairs in $\mathcal{G}_e$. In the graph OOD settings, the environments of the training and testing datasets, i.e. $\mathcal{E}_{tr}$ and $\mathcal{E}_{te}$ are always different, leading to the problem of distribution shift and the demand for model generalizability. The environment label $e$ for graphs is always unobserved since it is expensive to collect for most scenarios. Therefore, methods that impose requirements on environmental labels also limit their practical application.

Graph Invariant Learning (GIL) is aimed at learning an environment-agnostic function $f : \bigcup \mathcal{G} \to \mathcal{Y}$ to predict the label $Y$ for downstream tasks. The goal of GIL is to train an optimal $f^*$ that it generalizes well on all environments, we can formulate the problem as:

$$f^* = \arg\min_{f} \max_{e \in \mathcal{E}_{all}} \mathcal{R}(f|e), \tag{11}$$

where $\mathcal{R}(f|e) = \mathbb{E}_{(G,Y) \in \mathcal{G}^e}[l(f(G), Y)]$ is the risk of the predictor $f$ on the environment $e$ and $l : \mathcal{Y} \times \mathcal{Y} \to \mathbb{R}$ is the loss function.

The problem in Eq. 11 is difficult to solve since the environment variable $e$ is always unobserved. Recently, a new line of research has emerged that focuses on subgraphs, with the goal of identifying an invariant subgraph of the input, which has a stable relationship to the label; and filtering out the other part of the input, which is environment-relevant or spurious. These approaches are based on GNNs that aim to explicitly extract invariant subgraphs (with various definitions), guided by the information bottleneck (IB) principle (20; 10) or top-$k$ extraction based on causality (9; 11). In particular, (10) introduces Graph Stochastic Attention (GSAT), a novel attention mechanism that constructs inherently interpretable and generalizable GNNs. The attention is formulated as an information bottleneck by introducing stochasticity into the attention mechanism, which constrains the information flow from the input graph to the prediction. By penalizing the amount of information from the input data, GSAT is expected to be more generalizable. (11) proposes a Causality Inspired Invariant Graph LeArning (CIGA) framework to capture the invariance of graphs under various distribution shifts. Specifically, they characterize potential distribution shifts on graphs with causal models, which focus only on subgraphs containing the most information regarding the causes of labels. Overall, these new approaches provide exciting opportunities for achieving interpretability and generalizability in GNNs without requiring expensive domain labels.

### A.1.2 CARDINALITY-CONSTRAINED COMBINATORIAL OPTIMIZATION

Combinatorial optimization (CO) is a fundamental problem of computer science and operations research (49). Particularly, cardinality-constraint optimization is a permutation-based CO problem that exists widely in real-world applications (50; 51), whose final solution includes at most $k$ non-zero entries, i.e., the cardinality constraint $\|\mathbf{x}\|_0 \leq k$. Choosing the top-$k$ most influential edges for the invariant subgraph, which is a constraint-critical scenario, could also be regarded as a cardinality-constraint CO problem. Handling the constraint violation is the core of the cardinality-constraint CO problem as a tighter constraint violation leads to better performance (22). Erdos Goes Neural (52) places a penalty term for a constraint violation in the loss, but the constraint violation is unbounded. (26) develops a soft algorithm by recasting the top-$k$ selection as an optimal transport problem (53) with the Sinkhorn algorithm (24). Although the upper limit of constraint violation is provided, in the worst situation, the bound might diverge. (22) further addresses the issue and proposes a method with a tighter upper bound by introducing the Gumbel trick, making the constraint violation

arbitrarily controlled. These studies provide us with a fresh perspective and theoretical basis for invariance optimization for GIL in graph OOD.

## A.2 LEARNING OBJECTIVE DETAILS

Here we provide the detailed derivation of the lower bound of the mutual information term $I(G_S; Y)$ for our learning objective in Sec. 2.1, which includes a variational approximation distribution $P_\theta(Y|G_S)$ for tractability:

$$I(G_S; Y) = \iint_{G_S, Y} P(G_S, Y) \log \frac{P(G_S, Y)}{P(G_S)P(Y)} \dot{G}_S \dot{Y} \tag{12}$$

$$= \iint_{G_S, Y} P(G_S, Y) \log \frac{P(Y|G_S)}{P(Y)} \dot{G}_S \dot{Y} \tag{13}$$

$$= \iint_{G_S, Y} P(G_S, Y) \log P(Y|G_S) \dot{G}_S \dot{Y} \tag{14}$$

$$\quad - \iint_{G_S, Y} P(G_S, Y) \log P(Y) \dot{G}_S \dot{Y} \tag{15}$$

$$= \iint_{G_S, Y} P(G_S, Y) \log P(Y|G_S) \dot{G}_S \dot{Y} + H(Y) \tag{16}$$

$$= \iint_{G_S, Y} P(G_S) P(Y|G_S) \log P(Y|G_S) \dot{G}_S \dot{Y} + H(Y) \tag{17}$$

$$= \int_{G_S} P(G_S) \int_Y P(Y|G_S) \log P(Y|G_S) \dot{Y} \dot{G}_S + H(Y) \tag{18}$$

$$= \int_{G_S} P(G_S) \int_Y P(Y|G_S) \log \frac{P(Y|G_S) P_\theta(Y|G_S)}{P_\theta(Y|G_S)} \dot{Y} \dot{G}_S + H(Y) \tag{19}$$

$$= \int_{G_S} P(G_S) \left( \int_Y P(Y|G_S) \log P_\theta(Y|G_S) \dot{Y} + KL[P(Y|G_S) \| P_\theta(Y|G_S)] \right) \dot{G}_S \tag{20}$$

$$\quad + H(Y) \tag{21}$$

$$\geq \int_{G_S} P(G_S) \int_Y P(Y|G_S) \log P_\theta(Y|G_S) \dot{Y} \dot{G}_S + H(Y) \tag{22}$$

$$= \iint_{G_S, Y} P(G_S) P(Y|G_S) \log P_\theta(Y|G_S) \dot{Y} \dot{G}_S + H(Y) \tag{23}$$

$$= \iint_{G_S, Y} P(G_S, Y) \log P_\theta(Y|G_S) \dot{Y} \dot{G}_S + H(Y) \tag{24}$$

$$= \mathbb{E}_{G_S, Y} [\log P_\theta(Y|G_S)] + H(Y). \tag{25}$$

## A.3 SINKHORN ALGORITHM DETAILS

To solve the entropic regularized OT problem in Sec. 2.2:

$$\min_{\mathbf{T}} \text{tr}(\mathbf{T}^\top \mathbf{D}) - \tau H(\mathbf{T}), \text{ s.t. } \mathbf{T} \in [0, 1]^{2 \times N_e}, \ \mathbf{T1} = \mathbf{R}, \ \mathbf{T}^\top \mathbf{1} = \mathbf{C}, \tag{26}$$

where the discrete entropy item $H(\mathbf{T}) = \sum_{ij} -\mathbf{T}_{ij} (\log \mathbf{T}_{ij} - 1)$, by introducing Lagrangian multipliers $\boldsymbol{\alpha} \in \mathbb{R}^{2 \times 1}, \boldsymbol{\beta} \in \mathbb{R}^{N_e \times 1}$, the Lagrangian of the problem above is:

$$\min_{\mathbf{T}} \max_{\boldsymbol{\alpha}, \boldsymbol{\beta}} \mathcal{L}, \quad \text{where } \mathcal{L} = \text{tr}(\mathbf{T}^\top \mathbf{D}) - \tau H(\mathbf{T}) - \boldsymbol{\alpha}^\top (\mathbf{T1} - \mathbf{R}) - \boldsymbol{\beta}^\top (\mathbf{T}^\top \mathbf{1} - \mathbf{C}), \tag{27}$$

the first order conditions are:

$$\frac{\partial \mathcal{L}}{\partial \mathbf{T}} = \mathbf{D} + \tau \log \mathbf{T} - \boldsymbol{\alpha} \mathbf{1}_{1 \times N_e} - \mathbf{1}_{2 \times 1} \boldsymbol{\beta}^\top = 0, \tag{28}$$

$$\Leftrightarrow \mathbf{T} = \exp(-\frac{\mathbf{D} - \boldsymbol{\alpha} \mathbf{1}_{1 \times N_e} - \mathbf{1}_{2 \times 1} \boldsymbol{\beta}^\top}{\tau}), \tag{29}$$

$$\frac{\partial \mathcal{L}}{\partial \boldsymbol{\alpha}} = 0 \Leftrightarrow \mathbf{T}\mathbf{1} = \mathbf{R}, \quad \frac{\partial \mathcal{L}}{\partial \boldsymbol{\beta}} = 0 \Leftrightarrow \mathbf{T}^\top \mathbf{1} = \mathbf{C}, \tag{30}$$

the numerical analysis community provides an iterative solution as 'matrix scaling problem' (54):

$$\mathbf{T}_0 = \exp\left(-\frac{\mathbf{D}}{\tau}\right), \quad \mathbf{T}_k = \mathrm{diag}\left(\mathbf{T}_{k-1}\mathbf{1} \oslash \mathbf{R}\right)^{-1} \mathbf{T}_{k-1}, \quad \mathbf{T}_k = \mathbf{T}_{k-1} \mathrm{diag}(\mathbf{T}_{k-1}^\top \mathbf{1} \oslash \mathbf{C})^{-1}, \tag{31}$$

where $\mathbf{T}_0$ is the initialization, corresponding to the solution of $\frac{\partial \mathcal{L}}{\partial \mathbf{T}} = 0$ while $\boldsymbol{\alpha} = 0$ and $\boldsymbol{\beta} = 0$, the other two equations of $\mathbf{T}_k$ are iterations alternatively performed row- and column-wise normalizations corresponding to $\frac{\partial \mathcal{L}}{\partial \boldsymbol{\alpha}} = 0$ and $\frac{\partial \mathcal{L}}{\partial \boldsymbol{\beta}} = 0$, and $\oslash$ is element-wise division.

**Log-domain Sinkhorn.** For numerical stability, the Sinkhorn algorithm can be performed and implemented in the log domain to avoid the overflow problem caused by $\exp$ computations, instead of iterating $\mathbf{T}$, Log-domain Sinkhorn iterates $\log \mathbf{T}$:

$$\log \mathbf{T}_0 = -\frac{\mathbf{D}}{\tau}, \tag{32}$$

$$\log \mathbf{T}_k = \log \mathbf{T}_{k-1} - \log(\mathbf{T}_{k-1}\mathbf{1}) + \log \mathbf{R}, \tag{33}$$

$$\log \mathbf{T}_k = \log \mathbf{T}_{k-1} - \log(\mathbf{T}_{k-1}^\top \mathbf{1}) + \log \mathbf{C} = \log \mathbf{T}_{k-1} - \log(\mathbf{T}_{k-1}^\top \mathbf{1}), \tag{34}$$

where the items $\log(\mathbf{T}_{k-1}\mathbf{1})$ and $\log(\mathbf{T}_{k-1}^\top \mathbf{1})$ can be calculated by applying the 'logsumexp' operation on the first (row) and the second (column) dimension of $\log \mathbf{T}_{k-1}$. After the iterations of Log-domain Sinkhorn, the desired result $\mathbf{T}$ could be obtained by applying an $\exp$ operation on $\log \mathbf{T}$.

### A.4 TRAINING ALGORITHM DETAILS

Here we provide the procedure of GSINA training algorithm.

---

**Algorithm 1** The training procedure.

---

**Parameters**: the number of training epoch $E$; the number of batch size $B$; the sparsity $r$.
**Input**: training dataset $\mathcal{G} = \{G_i, Y_i\}_i^N$.
**Output**: the trained parameters $\theta$ and $\phi$ in Sec. 2.1.

1:  Initialize parameters $\theta$ and $\phi$;
2:  **for** $i = 1, \ldots, E$ **do**
3:      Sample data batches $\mathcal{B} = \{\mathcal{G}_1, \mathcal{G}_2, \ldots, \mathcal{G}_k\}$ from $\mathcal{G}$ with batch size $B$;
4:      **for** $j = 1, \ldots, k$ **do**
5:          $G = \{G_m | (G_m, Y_m) \in \mathcal{G}_j\}, Y = \{Y_m | (G_m, Y_m) \in \mathcal{G}_j\}$;
6:          Get inferred $G_S \sim g_\phi(G, r, \tau = 1)$ according to Eq. 9;
7:          Compute the gradients of the learning objective $\log P_\theta(Y | G_S)$ according to Eq. 3 and 10;
8:          Optimize parameters $\theta$ and $\phi$;
9:      **end for**
10: **end for**
11: Output the parameters $\theta$ and $\phi$;

---

## B  DATASET DETAILS

We follow the datasets used in the experiments of GSAT (10), CIGA (11) to test the generalizability of our GSINA on graph classification tasks, and datasets of EERM (15) on node classification tasks. Here we provide the details of these datasets in Tab. 7, 8, 9, 10.

Table 7: Statistics of graph classification datasets of GSAT benchmark used in Tab. 2, following the datasets used in DIR (9).

| | Spurious-Motif | | | MNIST-75sp (reduced) | | | Graph-SST2 | | | OBGB-Molhiv | | |
| | Train | Val | Test | Train | Val | Test | Train | Val | Test | Train | Val | Test |
|---|---|---|---|---|---|---|---|---|---|---|---|---|
| Classes# | | 3 | | | 10 | | | 2 | | | 2 | |
| Graphs# | 9,000 | 3,000 | 6,000 | 20,000 | 5,000 | 10,000 | 28,327 | 3,147 | 12,305 | 32,901 | 4,113 | 4,113 |
| Avg. N# | 25.4 | 26.1 | 88.7 | 66.8 | 67.3 | 67.0 | 17.7 | 17.3 | 3.45 | 25.3 | 27.79 | 25.3 |
| Avg. E# | 35.4 | 36.2 | 131.1 | 539.3 | 545.9 | 540.4 | 33.3 | 33.5 | 4.89 | 54.1 | 61.1 | 55.6 |
| Metrics | | ACC | | | ACC | | | ACC | | | ROC-AUC | |

Table 8: Summary of ogbg-mol* datasets of GSAT benchmark used for (multi-task) binary classification in Tab. 3. For all the datasets, we use the scaffold split with the split ratio of 80/10/10.

| Name | #Graphs | Average #Nodes | Average #Edges | #Tasks | Task Type | Metric |
|---|---|---|---|---|---|---|
| bace | 1,513 | 34.1 | 36.9 | 1 | Binary class. | ROC-AUC |
| bbbp | 2,039 | 24.1 | 26.0 | 1 | Binary class. | ROC-AUC |
| clintox | 1,477 | 26.2 | 27.9 | 2 | Binary class. | ROC-AUC |
| tox21 | 7,831 | 18.6 | 19.3 | 12 | Binary class. | ROC-AUC |
| sider | 1,427 | 33.6 | 35.4 | 27 | Binary class. | ROC-AUC |

Table 9: Statistics of the datasets of CIGA benchmark used in experiments.

| DATASETS | # TRAINING | # VALIDATION | # TESTING | # CLASSES | # NODES | # EDGES | METRICS |
|---|---|---|---|---|---|---|---|
| SPMOTIF | 9,000 | 3,000 | 3,000 | 3 | 44.96 | 65.67 | ACC |
| SST5 | 6,090 | 1,186 | 2,240 | 5 | 19.85 | 37.70 | ACC |
| TWITTER | 3,238 | 694 | 1,509 | 3 | 21.10 | 40.20 | ACC |
| DRUGOOD-ASSAY | 34,179 | 19,028 | 19,032 | 2 | 32.27 | 70.25 | ROC-AUC |
| DRUGOOD-SCAFFOLD | 21,519 | 19,041 | 19,048 | 2 | 29.95 | 64.86 | ROC-AUC |
| DRUGOOD-SIZE | 36,597 | 17,660 | 16,415 | 2 | 30.73 | 66.90 | ROC-AUC |
| PROTEINS | 511 | 56 | 112 | 2 | 39.06 | 145.63 | MCC |
| DD | 533 | 59 | 118 | 2 | 284.32 | 1,431.32 | MCC |
| NCI1 | 1,942 | 215 | 412 | 2 | 29.87 | 64.6 | MCC |
| NCI109 | 1,872 | 207 | 421 | 2 | 29.68 | 64.26 | MCC |

Table 10: Statistics of the datasets of EERM benchmark used in experiments.

| Dataset | Distribution Shift | #Nodes | #Edges | #Classes | Train/Val/Test Split | Metric | Adapted From |
|---|---|---|---|---|---|---|---|
| Cora | Artificial Transformation | 2,703 | 5,278 | 10 | by graphs | Accuracy | (55) |
| Amazon-Photo | | 7,650 | 119,081 | 10 | by graphs | Accuracy | (56) |
| Twitch-explicit | Cross-Domain Transfers | 1,912 - 9,498 | 31,299 - 153,138 | 2 | by graphs | ROC-AUC | (57) |
| Facebook-100 | | 769 - 41,536 | 16,656 - 1,590,655 | 2 | by graphs | Accuracy | (58) |
| Elliptic | Temporal Evolution | 203,769 | 234,355 | 2 | by time | F1 Score | (59)[1] |
| OGB-Arxiv | | 169,343 | 1,166,243 | 40 | by time | Accuracy | (60) |

# C IMPLEMENTATION DETAILS

## C.1 GNN BACKBONES

For GSINA with GIN and PNA backbones for the experiments on GSAT benchmark (reported in Tab. 2, 3), our GNN backbone settings of GIN and PNA are strictly in line with those in GSAT settings. We use 2 layers GIN with 64 hidden dimensions and 0.3 dropout ratio. We use 4 layers PNA with 80 hidden dimensions, 0.3 dropout ratio, and no scalars are used. We directly follow PNA and GSAT using (mean, min, max, std) aggregators for OGBG datasets, and (mean, min, max, std, sum) aggregators for all other datasets.

For GSINA with GCN or GIN layers for the experiments on CIGA benchmark (reported in Tab. 4, 5), our GNN backbone settings are also strictly in line with those in CIGA settings. We use 3-layer GNN with Batch Normalization between layers and JK residual connections at last layer. We use GCN with mean readout for all datasets except Proteins and DrugOOD datasets. For Proteins, we use GIN and max readout. For DrugOOD datasets, we use 4-layer GIN with sum readout. The hidden dimensions are fixed as 32 for SPMotif, TU datasets, and 128 for SST5, Twitter, and DrugOOD datasets.

For GSINA with GCN and SAGE backbones for the experiments on EERM benchmark (reported in Fig. 3, 5, 6), our GNN backbone settings of GCN and SAGE are strictly in line with those in EERM settings, where we use ReLU as the activation, add self-loops and use batch normalization for graph convolution in each layer. We use 2 layers GCN with hidden size 32 for Cora, Amazon-Photo, Twitch-explicit, and Facebook-100; and 5 layers SAGE with hidden size 32 for Elliptic and OGB-Arxiv.

## C.2 TRAINING DETAILS

All experiments are conducted for 5 runs on RTX-2080Ti (11GB) GPUs, and the average and standard deviation are reported.

**Optimization.** We use Adam optimizers for graph classification experiments reported in Tab. 2, 3, 4, 5 strictly in line with the settings of GSAT and CIGA. For the experiments on the GSAT benchmark, our GSINA with GIN backbone uses 0.003 learning rate for Spurious-Motifs and 0.001 for all other datasets. Our GSINA with PNA backbone uses 0.003 learning rate for Graph-SST2 and Spurious-Motifs, and 0.001 learning rate for all other datasets. For the experiments on the CIGA benchmark, we use 0.001 learning rate for all datasets. For the experiments on the EERM benchmark, for all datasets, we follow the AdamW optimizers used in EERM. To reproduce EERM results, we strictly follow the original settings of EERM. For the setting of our GSINA, we use the same settings with the experiments of ERM, where we use 0.01 learning rate for Cora, Amazon-Photo, and Twitch-explicit, 0.001 for Facebook-100, and 0.0002 for Elliptic.

**Batch Size.** For the experiments on the GSAT benchmark, we use a batch size of 128 for all datasets. For the experiments on the CIGA benchmark, strictly in line with CIGA, we use a batch size of 32 for all datasets, except for DrugOOD datasets, where we use 128.

**Epoch.** For graph classification experiments on GSAT and CIGA benchmarks, we perform early stopping to avoid overfitting. Based on the difficulty of fitting the dataset, we set the early stopping patience to 10 for the SPMotif (both GSAT and CIGA benchmarks) datasets, DrugOOD-Size and TU datasets, 3 for Graph-SST2 and Graph-SST5, 5 for Twitter, 20 for DrugOOD-Assay/Scaffold and 30 for MNIST-75sp. We do not use early stopping for OGBG datasets (molhiv, molbace, molbbbp, molclintox, moltox21, molsider), and simply train them to the end of epochs (200 epochs for GIN + GSINA, 100 for PNA + GSINA) to achieve better performances. For TU datasets, like the practices of pre-training in CIGA, we pretrain them for 30 epochs to avoid underfitting. For EERM benchmark, we do not perform early stoppings or pre-trainings, we follow the epochs used in EERM work and train to the end, which is 200 epochs for all datasets except for 500 epochs for OGB-Arxiv.

## C.3 BASELINES

The baseline settings of GSAT, CIGA, other interpretable GNNs (ASAP, GIB, and DIR), EERM, and invariant learning methods (ERM, IRM, V-Rex, EIIL, IB-IRM, CNC) are strictly in line with the settings reported in GSAT, CIGA, and EERM, and we strictly cite their experimental results for fair comparisons.

# D MORE EXPERIMENTS AND EXPERIMENTAL DETAILS

## D.1 MODEL SELECTION

Fig. 7, 8 provide the validation performances of GSINA on GSAT benchmark (experiments in Tab. 2, 3), which guide the selection of the sparsity hyperparameter $r$ in our GIL framework GSINA.

**Hyperparameter Setting of Sparsity $r$.** As described in Sec. 3, we perform $r$ selection on the GSAT benchmark based on the validation performances. For GIN backboned GSINA, we set $r = 0.9$ for OGBG-Molhiv, 0.2 for Graph-SST2, 0.5 for MNIST-75sp, 0.6 for SPMotif-0.5, 0.6 for SPMotif-0.7, and 0.4 for SPMotif-0.9. For PNA backboned GSINA, we set $r = 0.7$ for OGBG-Molhiv, 0.5 for OGBG-Molbace, 0.8 for OGBG-Molbbbp, 0.7 for OGBG-Molclintox, 0.7 for OGBG-Moltox21, 0.8 for OGBG-Molsider, 0.8 for Graph-SST2, 0.6 for MNIST-75sp, 0.1 for SPMotif-0.5, 0.3 for SPMotif-0.7, and 0.5 for SPMotif-0.9.

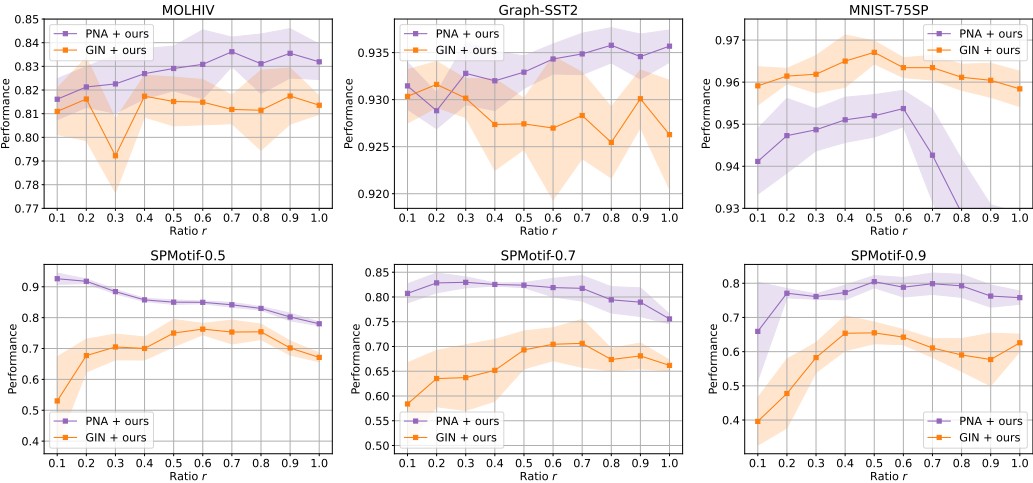

Figure 7: Validation performances for different $r$ in GSINA.

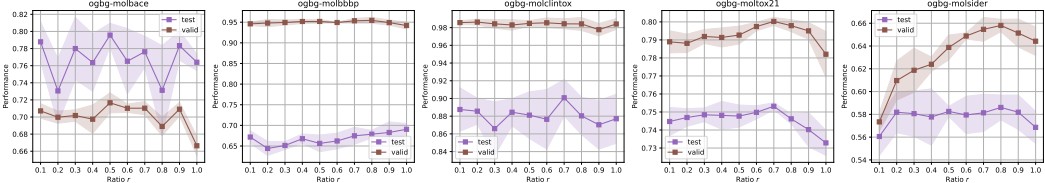

Figure 8: Validation as well as testing performances for different $r$ on OGBG-Mol* datasets.

For the experiments on the CIGA benchmark, as CIGA also performs top-$r$ selection and provides its setting of sparsity hyperparameter $r$, we strictly follow its setting of $r$, which are 0.25 for SPMotif, 0.3 for Proteins and DD, 0.6 for NCI1, 0.7 for NCI109, 0.5 for SST5 and Twitter, and 0.8 for DrugOOD datasets, respectively.

For the experiments on the EERM benchmark, as described in Sec. 3, we uniformly set $r = 0.5$.

**Batch Computing and Complexity analysis.** Our GSINA has two different settings for subgraph extraction in a batch (we selected the better one for experiments). The first (denoted as micro) setting is to compute soft top-$r$ for each graph in the batch, which is slower, we use it on Spmotif, TU-PROT, TU-DD, Graph-SST2, MNIST-75sp, OGBG-Molbbbp, OGBG-Molclintox, OGBG-Moltox21, OGBG-Molsider. The second (denoted as macro) setting is to compute soft top-r just once for the entire batch (which can be considered as a large graph composed of several smaller graphs), which is faster, we use it on Graph-SST5, Twitter, Drug-Assay, Drug-Sca, Drug-Size, TU-NCI1, TU-NCI109, OGBG-Molhiv, OGBG-Molbace. The macro setting computes the top-$r$ for the whole graph and relaxes the constraint of top-$r$ on each individual graph, which is useful when the distribution shifts are complicated, and in that case, it is difficult to find reasonable $r$ for each graph. Therefore, the macro vs micro model selection in GSINA is highly dependent on the characteristics of the dataset, and we select the one with better performance for each dataset. On the other hand, CIGA uses hard top-k selection for subgraph extraction for each graph (similar to the micro setting) in the batch. And we have implemented GSAT (without hyperparameter tuning) on the CIGA benchmark for training time comparison, GSAT is similar to the macro setting, GSAT models the probability $p$ of each edge belonging to the invariant part, which does not consider how many graphs are in the batch.

We conducted experiments on the training time of the models on all datasets of the CIGA benchmark (Statistics of the benchmark datasets can be found in Appendix B). In Tab. 11, we report experimental results of training time comparison on CIGA benchmark, ERM is always the fastest, and our second setting (GSINA-macro) is often much faster than CIGA and a bit slower than GSAT due to iterative computations (Eq. 6); while the first setting (GSINA-micro), also due to iterative computations (Eq. 6) in subgraph extraction, is slower than CIGA's hard top-k selection. However, since we

Table 11: Training time comparison on CIGA benchmark, we record the mean ± std training time (in terms of ms/epoch).

| | ERM | GSAT | GSINA-macro | CIGA | GSINA-micro |
|---|---|---|---|---|---|
| Spmotif | 5797.0464 ± 215.3673 | 11084.8443 ± 290.5553 | 14799.7564 ± 315.1995 | 21658.0957 ± 908.5341 | 34576.2656 ± 628.3021 |
| Graph-SST5 | 3993.3391 ± 99.6189 | 7737.8888 ± 84.9996 | 9503.5158 ± 234.1883 | 14344.7820 ± 246.2677 | 24521.2875 ± 369.2448 |
| Twitter | 2123.6215 ± 289.1412 | 4129.1175 ± 138.3962 | 5214.8427 ± 125.0164 | 7958.9061 ± 208.1967 | 13038.0312 ± 214.8203 |
| Drug-Assay | 9771.7190 ± 2122.2238 | 17026.1180 ± 657.9675 | 19833.7992 ± 1240.6085 | 54833.6117 ± 1221.3473 | 97431.7109 ± 3121.8496 |
| Drug-Sca | 6704.2010 ± 658.8927 | 10845.3604 ± 597.7060 | 12675.0961 ± 670.9245 | 33741.7516 ± 615.7502 | 59474.7531 ± 689.3121 |
| Drug-Size | 10814.6467 ± 1104.6025 | 17952.6367 ± 1071.1815 | 23152.3664 ± 1540.6074 | 58453.0594 ± 924.8912 | 101284.0609 ± 751.5329 |
| TU-NCI1 | 1258.8495 ± 71.3337 | 2450.3426 ± 98.2116 | 3141.5663 ± 116.0735 | 5014.3386 ± 126.6357 | 7429.4210 ± 113.3016 |
| TU-NCI109 | 1217.0766 ± 77.9733 | 2394.2303 ± 81.2618 | 2995.5683 ± 90.5246 | 4481.9476 ± 54.5312 | 7373.4569 ± 125.3487 |
| TU-PROT | 400.4288 ± 80.5055 | 714.7085 ± 79.7937 | 891.0935 ± 74.9539 | 1190.0512 ± 82.4018 | 1929.6349 ± 94.4660 |
| TU-DD | 369.0103 ± 92.8588 | 717.2253 ± 89.7630 | 1035.9521 ± 63.5705 | 1361.7259 ± 113.0401 | 2141.5195 ± 109.6949 |

can set the number of iterations to a relatively small value (we uniformly used 10), GSINA does not introduce a significant computing overhead and is always within $2 \times$ CIGA's complexity.

**More Settings.** In our GSINA framework, we fix the Sinkhorn temperature $\tau$ (for softness) and Gumbel noise factor $\sigma$ (for randomness) to 1, and the Sinkhorn iteration numbers to 10. As shown in Fig. 9, we analyze the hyperparameter sensitivity of the Sinkhorn temperature $\tau \in \{0.2, 0.5, 1, 2, 5\}$ on the SPMotif datasets (of GSAT benchmark, with both GIN and PNA backboned GSINA), which demonstrates $\tau = 1$ is a reasonable and effective choice. When $\tau$ is too small, the model lacks softness and generates too 'hard' subgraphs; when $\tau$ is too large, the model is too soft and generates subgraphs that are too smooth and lacks information. Both cases lead to reduced performances.

Moreover, we demonstrate the expressiveness of the settings in Fig. 10, where Fig. 10a is randomly generated edge scores (we sort them by values, each row is a graph and each column is an edge score) to mimic the distributions of the edge scores output from $\text{MLP}_\phi$, the bottom lower region is non-edge padding; Fig. 10b is the ground truth of top-$r$ ($r = 0.3$) on the edge scores in Fig. 10a; Fig. 10c, 10d are edge attention values output from GSINA ($r = 0.3$) computations, for 1 run result and 10 runs average, respectively. Fig. 10 demonstrate that the settings are sensitive to the edge scores output from $\text{MLP}_\phi$, and could generate distinguishable GSINA edge attention results, thus expressive enough for the learning of GSA.

The $\text{MLP}_\phi$ in our subgraph extractor $g_\phi$ is a 2-layer MLP, $\text{MLP}_\phi(\mathbf{h}_i, \mathbf{h}_j)$ is implemented by inputting concatenated $[\mathbf{h}_i, \mathbf{h}_j]$, and outputting a 1-dim edge score $s_{ij}$, then a batch normalization layer ($s := \frac{s - \text{mean}(s)}{\text{std}(s)}$) is applied to normalize the edge scores to a stable value range.

The aggregator of GSINA node attention to aggregate the edge attention values in node neighborhood is uniformly set as the 'max' aggregator for all cases.

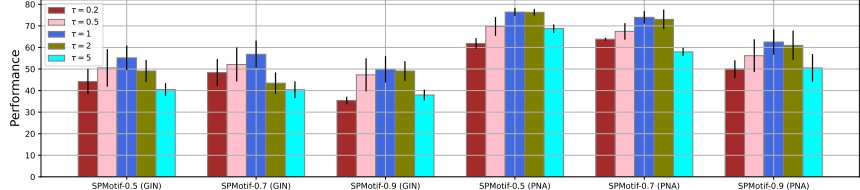

Figure 9: Performances of different Sinkhorn temperature $\tau$ (for softness) on SPMotif datasets.

**More Discussions.** Our GSINA's model selection procedure is simpler than CIGA (the 'hard' top-$k$ based GIL SOTA), with other settings fixed, what is required to tune to achieve SOTA performances in GSINA is just the sparsity $r$, while CIGA has additional loss balancing hyperparameters (2 losses corresponding to CIGAv1 and CIGAv2) to tune. Besides, the 'hard' top-$k$ methods could not be utilized to improve node-level tasks, as part of nodes would be simply discarded and their representations naturally could not be learned in 'hard' top-$k$ methods (e.g. DIR, CIGA).

### D.2 INTERPRETATION PERFORMANCES AND FURTHER DISCUSSIONS

In addition to generalizability detailedly discussed in our paper, here we report the interpretability of our GSINA. The interpretability is evaluated based on SPMotif datasets, which have labeled ground truths of explanation subgraphs (a.k.a invariant subgraphs $G_S$). The interpretability evaluation metrics are following previous studies DIR (9) and GSAT (10) for the performance of explanation subgraph recognition, which is a problem of binary classification for each edge. Following DIR and GSAT, we perform edge binary classifications, for metrics, we evaluate ROC-AUC (with GIN

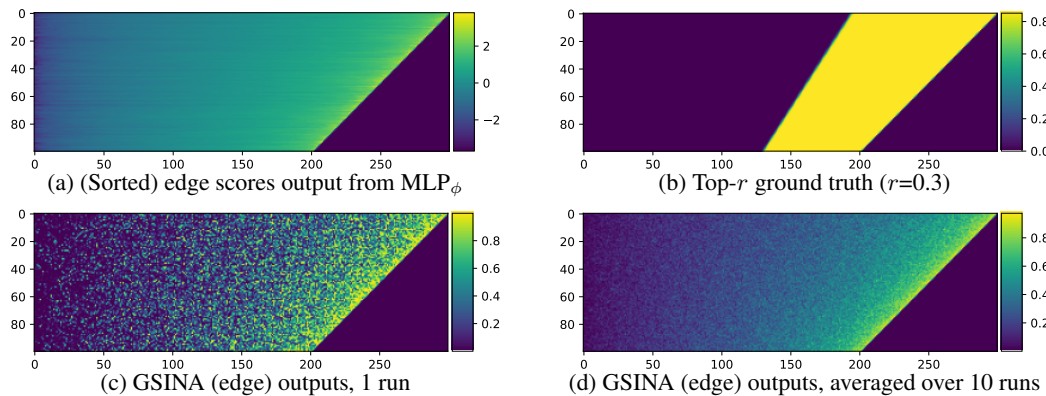

Figure 10: Demonstration of the expressiveness of the fixed settings in our GSINA.

(a) (Sorted) edge scores output from $\text{MLP}_\phi$

(b) Top-$r$ ground truth ($r=0.3$)

(c) GSINA (edge) outputs, 1 run

(d) GSINA (edge) outputs, averaged over 10 runs

Table 12: Interpretation Performance (AUC).

| | SPURIOUS-MOTIF | | |
| | $b = 0.5$ | $b = 0.7$ | $b = 0.9$ |
| --- | --- | --- | --- |
| GNNEXPLAINER | $62.62_{\pm 1.35}$ | $62.25_{\pm 3.61}$ | $58.86_{\pm 1.93}$ |
| PGEXPLAINER | $69.54_{\pm 5.64}$ | $72.33_{\pm 9.18}$ | $\underline{72.34}_{\pm 2.91}$ |
| GRAPHMASK | $72.06_{\pm 5.58}$ | $73.06_{\pm 4.91}$ | $66.68_{\pm 6.96}$ |
| GIB (20) | $57.29_{\pm 14.35}$ | $62.89_{\pm 15.59}$ | $47.29_{\pm 13.39}$ |
| DIR (9) | $\underline{78.15}_{\pm 1.32}$ | $\underline{77.68}_{\pm 1.22}$ | $49.08_{\pm 3.66}$ |
| GIN+GSAT | $\mathbf{78.45}_{\pm 3.12}$ | $74.07_{\pm 5.28}$ | $71.97_{\pm 4.41}$ |
| GIN+OURS | $65.13_{\pm 10.00}$ | $60.78_{\pm 8.09}$ | $55.86_{\pm 3.56}$ |
| PNA+GSAT | $\mathbf{83.34}_{\pm 2.17}$ | $\mathbf{86.94}_{\pm 4.05}$ | $\mathbf{88.66}_{\pm 2.44}$ |
| PNA+OURS | $75.66_{\pm 1.56}$ | $80.47_{\pm 1.06}$ | $80.10_{\pm 2.04}$ |

Table 13: Interpretation precision@5 of with the GNN backbone 'SPMotifNet' in DIR (9).

| | SPURIOUS-MOTIF | | |
| | $b = 0.5$ | $b = 0.7$ | $b = 0.9$ |
| --- | --- | --- | --- |
| GNNEXPLAINER | $0.203_{\pm 0.019}$ | $0.167_{\pm 0.039}$ | $0.066_{\pm 0.007}$ |
| DIR (9) | $0.255_{\pm 0.016}$ | $0.247_{\pm 0.012}$ | $0.192_{\pm 0.044}$ |
| GSAT (10) | $\mathbf{0.519}_{\pm 0.022}$ | $\mathbf{0.503}_{\pm 0.034}$ | $0.416_{\pm 0.081}$ |
| OURS | $0.419_{\pm 0.030}$ | $0.401_{\pm 0.046}$ | $\mathbf{0.429}_{\pm 0.040}$ |

and PNA backbones, following the setting of GSAT) and precision@5 (with the GNN backbone 'SPMotifNet' used in DIR (9)).

As shown in Tab. 12, 13, our GSINA (PNA and SPMotifNet backbones) mostly outperform interpretable GNNs DIR and GIB as well as other post-hoc (10) GNN explainers, showing our inherent interpretability to extract invariant subgraphs. However, our GSINA is inferior to GSAT in interpretability, we tend to believe that it is due to the innate characteristics of top-$k$ based methods: in the 'hard' top-$k$ case, the output value is binary (0 / 1), meaning an item belongs to top-$k$ or not, 'hard' top-$k$ does not consider the relative ranking of items at all, while the prevalent metrics for binary classification (e.g. AUC, precision@5) always consider. Hence, it is not natural to evaluate the subgraph recognition performance of top-$k$ based methods. However, GSAT does not restrict edge predictions to (approximate) binary like top-$k$ methods, so that it achieves better performance in interpretability. Due to the utilization of soft top-$r$ operation in GSINA, it places more mathematical constraints on the graph attention value distribution compared to GSAT. The attention of GSAT is more flexible because it calculates the probability of each edge belonging to the invariant part separately, without considering the global constraint on the ratio ($r$) of the invariant subgraph. The choice of r in GSINA significantly impacts the attention distribution, whereas GSAT is not subject to this issue, which is why GSAT's attention distribution is more flexible. Furthermore, The design of GSAT, as mentioned above, is a tradeoff. It excels in interpretability, but it falls short in effectively

filtering out variant parts (as described in Sec. 1, Fig. 1). This leads to GSAT having a less optimal ability to make predictions using subgraphs compared to GSINA.

## D.3 INTERPRETATION VISUALIZATION

Here we visualize the subgraph extractions on several datasets in GSAT benchmark. We visualize the original edge and node attention values given by our GSA, and do not perform the scaling tricks: first normalization, then multiplying edges' numerical values repeatedly (10 times in GSAT) to improve discrimination, which is applied in visualizations of GSAT and CIGA.

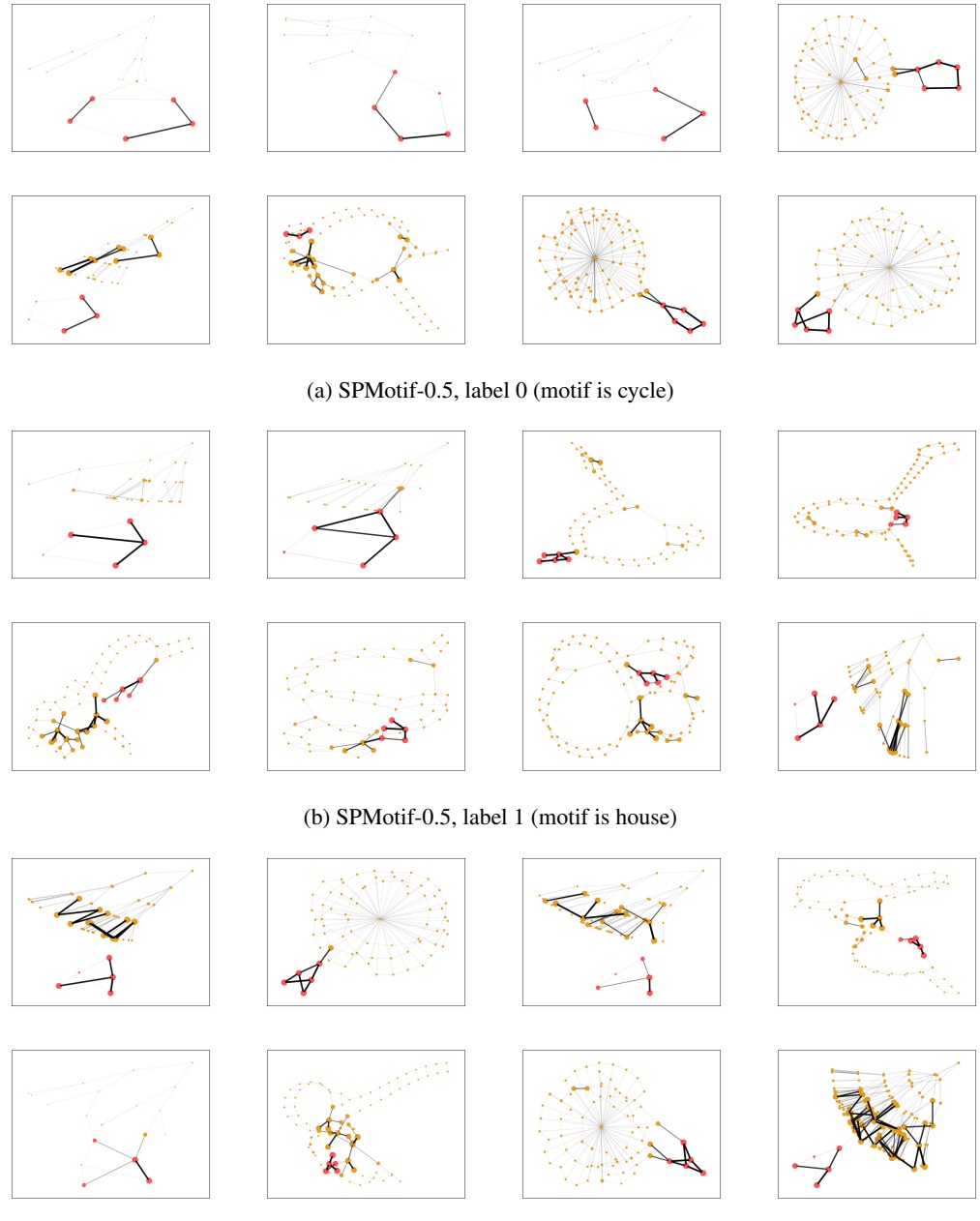

(a) SPMotif-0.5, label 0 (motif is cycle)

(b) SPMotif-0.5, label 1 (motif is house)

(c) SPMotif-0.5, label 2 (motif is crane)

Figure 11: Visualization of the extracted subgraphs in SPMotif-0.5 dataset.

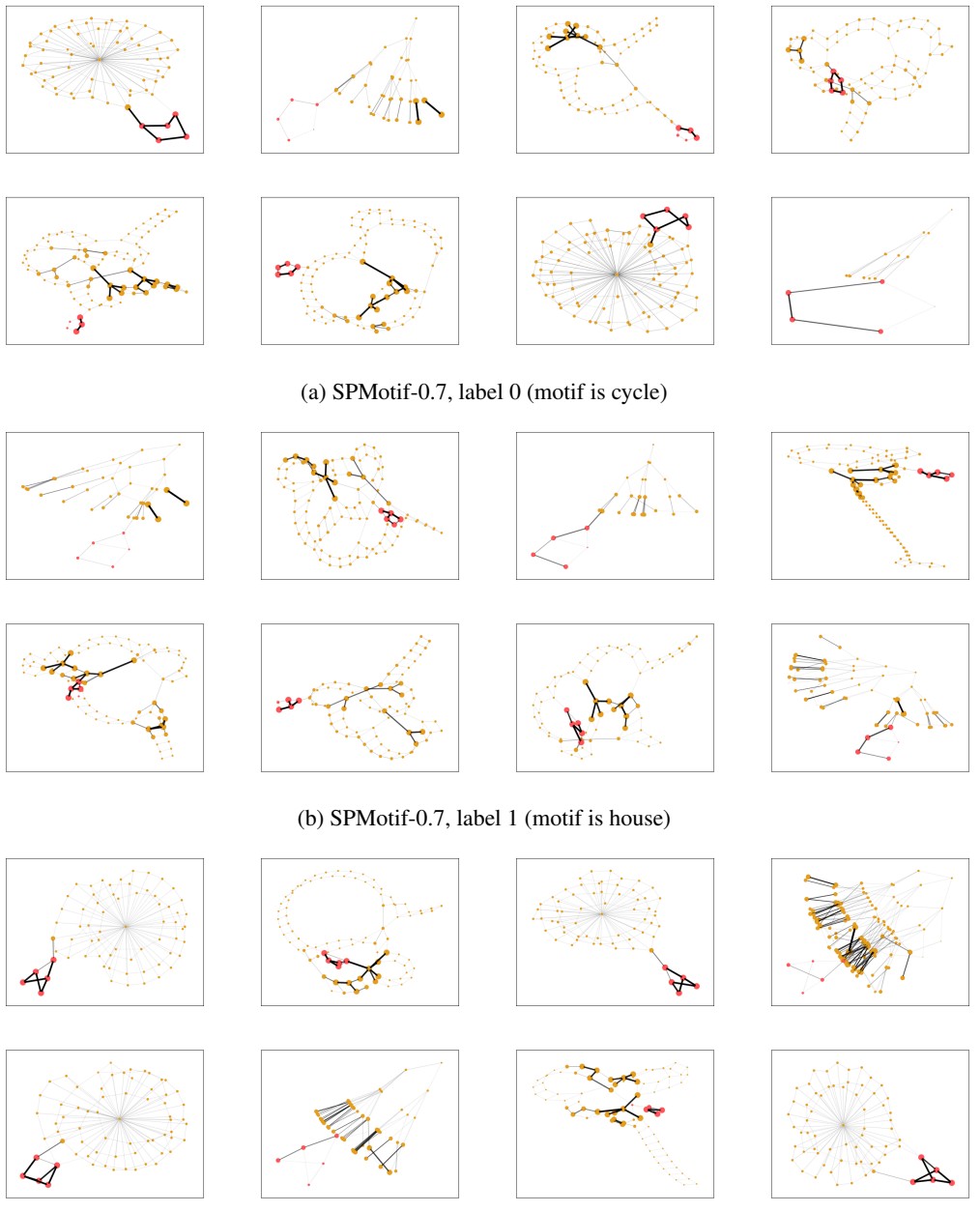

(a) SPMotif-0.7, label 0 (motif is cycle)

(b) SPMotif-0.7, label 1 (motif is house)

(c) SPMotif-0.7, label 2 (motif is crane)

Figure 12: Visualization of the extracted subgraphs in SPMotif-0.7 dataset.

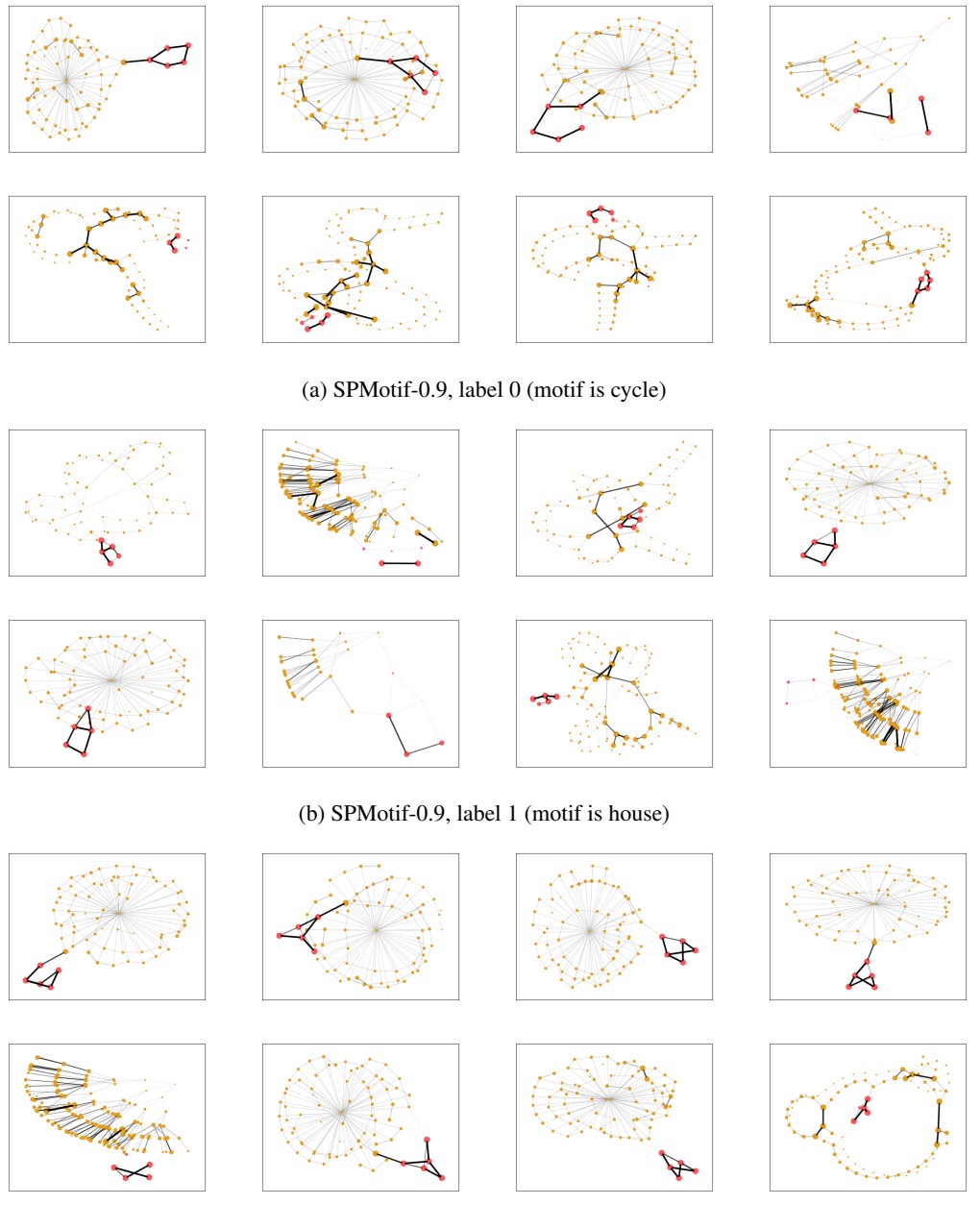

(a) SPMotif-0.9, label 0 (motif is cycle)

(b) SPMotif-0.9, label 1 (motif is house)

(c) SPMotif-0.9, label 2 (motif is crane)

Figure 13: Visualization of the extracted subgraphs in SPMotif-0.9 dataset.

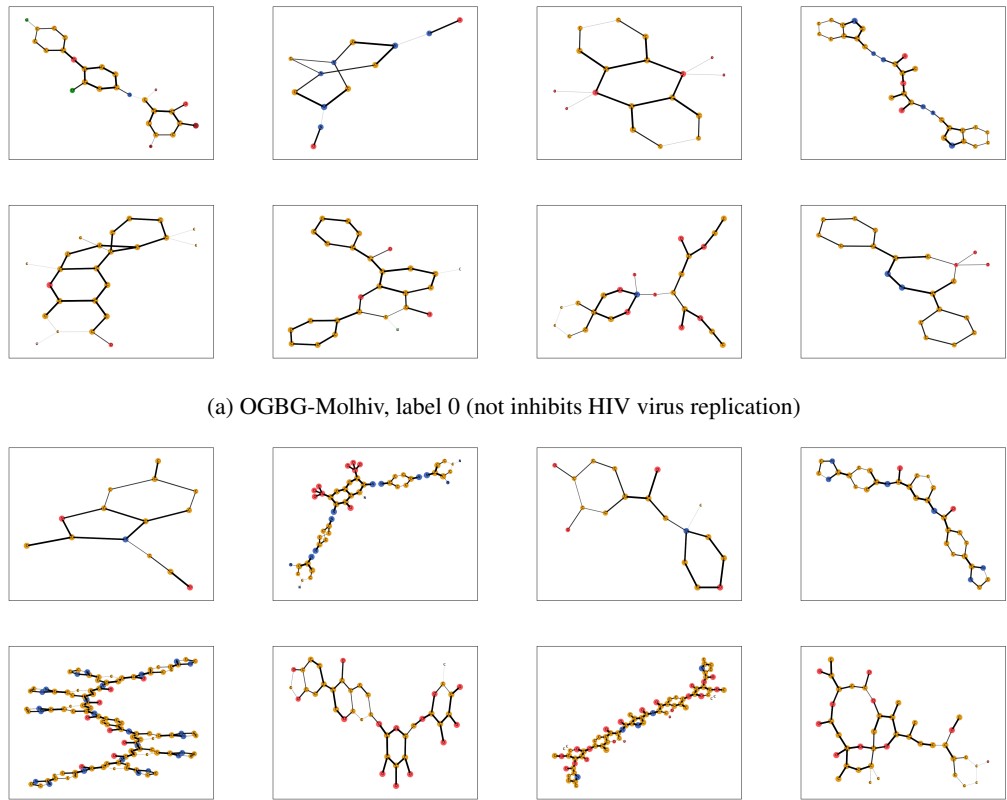

(a) OGBG-Molhiv, label 0 (not inhibits HIV virus replication)

(b) OGBG-Molhiv, label 1 (inhibits HIV virus replication)

Figure 14: Visualization of the extracted subgraphs in OGBG-Molhiv dataset.

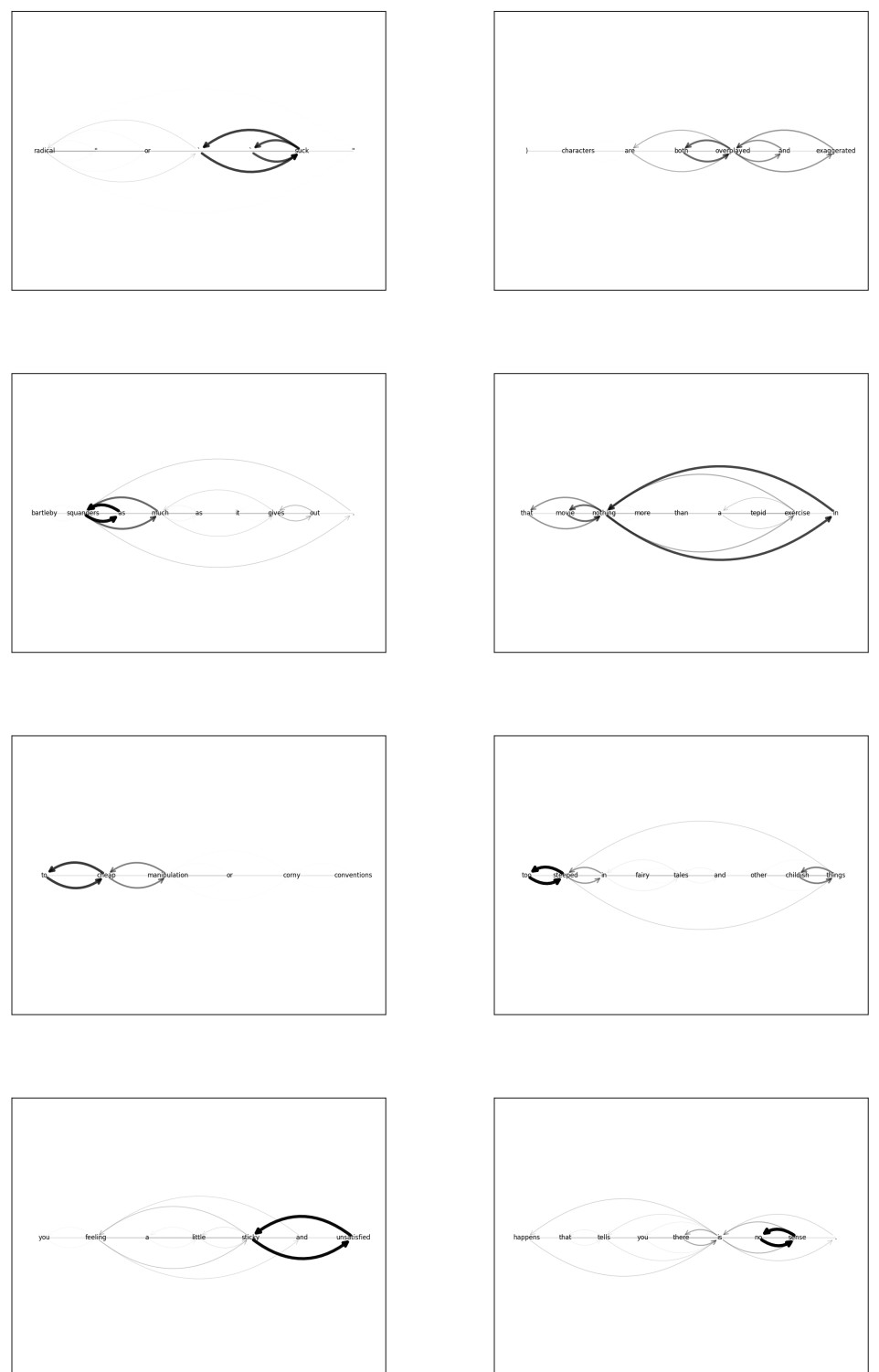

Figure 15: Graph-SST2, label 0 (negative sentiment)

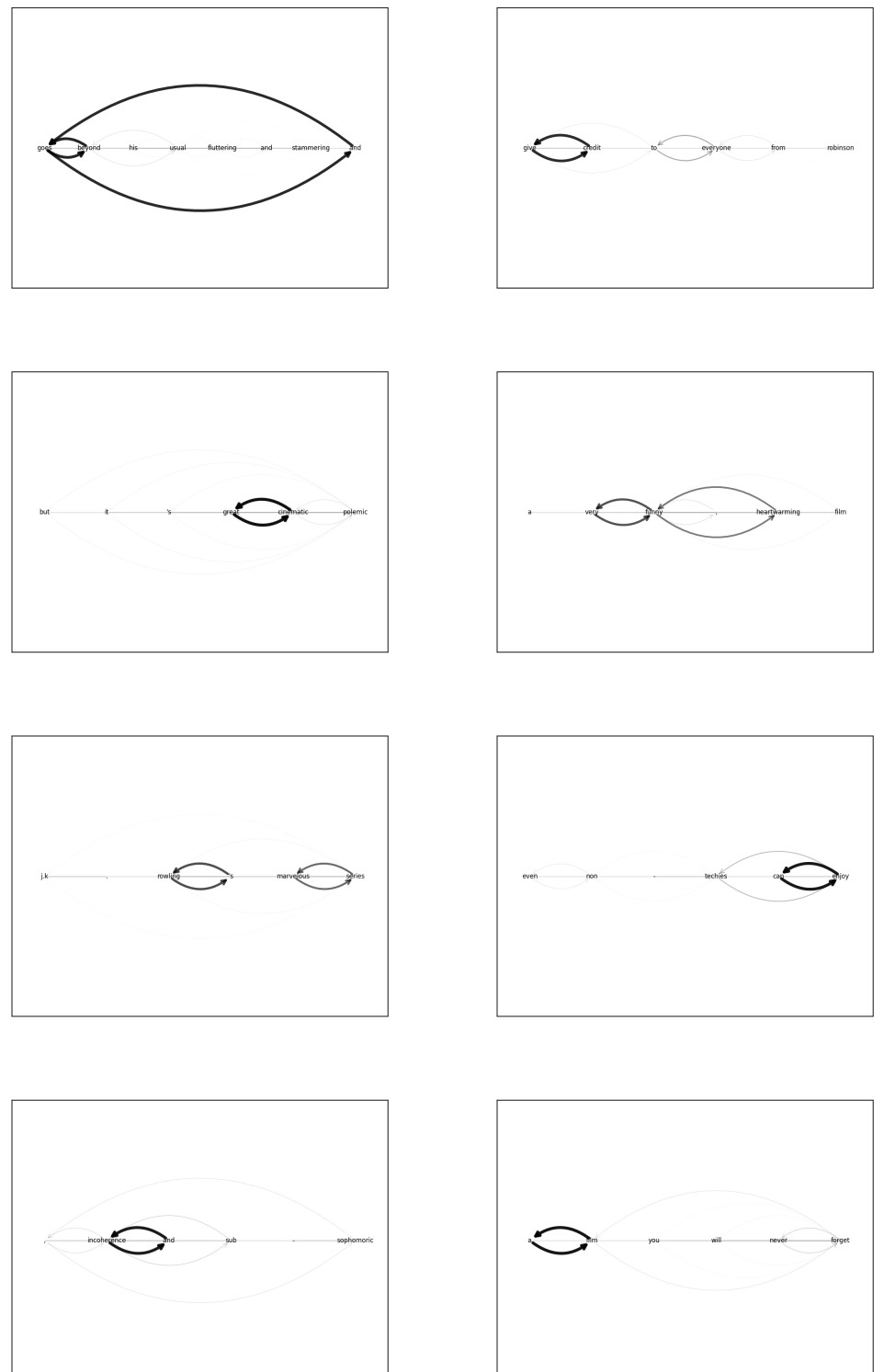

Figure 16: Graph-SST2, label 1 (positive sentiment)

