# OpenReview forum: "GSINA: Improving Graph Invariant Learning via Graph Sinkhorn Attention"
_ICLR.cc/2024/Conference — ICLR 2024 Conference Withdrawn Submission_

### Official Review · Reviewer_VjfH · 2023-10-31

**Soundness:** 2 fair
**Presentation:** 2 fair
**Contribution:** 2 fair
**Rating:** 5
**Confidence:** 4

**Summary:**

This paper studies the problem of graph invariant learning (GIL), aiming to find edges or nodes that are related to label information and invariant to environmental changes. This paper proposes three principles in GIL: Sparsity, Softness, and Differentiability, which cannot be fully covered by previous GIL methods. To address this issue, this paper designs a new regularization, namely Graph Sinkhorn Attention (GSINA), based on the optimal transport theory. GSINA can control the sparsity and softness of edge attention, and therefore improve the performance of GIL. Experiments on both real and synthetic datasets validate the effectiveness of the proposed method.

**Strengths:**

1. The idea of utilizing the GIL for optimal transport is somewhat interesting. It makes sense to move the coefficients to 1 for invariant edges and to 0 for spurious edges.

2. The proposed method GSINA consistently outperforms other GIL methods.

**Weaknesses:**

1. In addition to the three principles proposed, I think there is a fourth principle, which is the completeness of the subgraph. In practice, we expect important edges to form a complete subgraph. For example, in molecular property prediction, invariant information should be related to functional groups rather than individual chemical bonds or atoms. This is the advantage of subgraph selection methods over information bottleneck methods. I am concerned about whether enforcing sparsity guarantees this principle.

2. There seems to be no clear reason to replace information bottleneck with optimal transport. We can also apply the Gumbel trick to  Graph Stochastic Attention (GSAT) to ensure its sparsity. Is there any theoretical evidence to prove the effectiveness of optimal transport over information bottleneck? Additionally, we can observe from the ablation study (Table 6) that without the help of the Gumbel trick, GSINA performs similarly or even worse than GSAT. Therefore, I am concerned about the effectiveness of the optimal transport.

3. The experimental results appear to be copied from other papers. But I think it would be better if this paper could provide the results for some important baselines. For example, GSAT in the graph-level OOD tasks.

**Questions:**

See weaknesses.

---

> ### Author Response · Authors · 2023-11-13
>
> Thank you for your review. We believe that there may be some misunderstandings, and we would like to address your comments point by point:
>
> **1. Regarding the issue of subgraph completeness: as you mentioned, a complete subgraph may be beneficial for property prediction compared to an incomplete subgraph (according to human understanding). However, guaranteeing completeness is a challenging problem due to the following reasons:**
>
>    - The invariant subgraph is latent and inferred by the GNN from the data. The model's understanding of the data may not align with human understanding. For example, in Appendix D Fig.12 (a), the invariant subgraph inferred by GSINA may be discontinuous (incomplete), but the model still considers such subgraphs beneficial for improving GIL performance. On the other hand, what humans consider a “complete subgraph” may still have redundant parts in the model's learning process. It is difficult for us to determine definitively which nodes or edges should be included in a “complete invariant subgraph”.
>
>    - Since many datasets may lack ground-truth information about the invariant subgraph, it is also challenging to define a metric or penalty loss for completeness.
>
>    - If we consider the issue of connectivity (different from completeness), you can refer to the reply to reviewer 8Epa's 4th point in https://openreview.net/forum?id=MfWFUJklRI&noteId=zBciutJrKy. We can add a connectivity penalty loss to our learning objective in Sec. 2.1, but it may not necessarily have a significant impact because the requirement for connectivity in the invariant subgraph depends on the data characteristics.
>
>    - Lastly, sparsity is not closely related to completeness. Sparsity refers to the ratio of the invariant subgraph to the entire graph.
>
> **2. Regarding the effectiveness of GSINA, there might be some misunderstandings. We would like to clarify them as follows:**
>
>    - The reason for replacing the information bottleneck with OT-theoretic optimal transport (Eq. 5, 6) is already mentioned in our Sec. 1 analysis. The effectiveness of information bottleneck-based methods (GSAT, GIB, etc.) for spurious feature filtering is limited and does not guarantee subgraph sparsity. For example, GSAT models the probability $p$ of each edge belonging to the invariant part, and the "sparsity" of GSAT depends on whether this prior probability is set small enough. However, as mentioned in the GSAT repository https://github.com/Graph-COM/GSAT  (quoted below), GSAT has limitations in sparsity, and it is also difficult to constrain the final edge attention distribution by the prior.
>
> >```
> >Does GSAT encourage sparsity?
> >No, GSAT doesn't encourage generating sparse subgraphs. We find r = 0.7 (Eq.(9) in our paper) can generally work well for all datasets in our experiments, which means during training roughly 70% of edges will be kept (kind of still large). This is because GSAT doesn't try to provide interpretability by finding a small/sparse subgraph of the original input graph, which is what previous works normally do and will hurt performance significantly for inhrently interpretable models (as shown in Fig. 7 in the paper). By contrast, GSAT provides interpretability by pushing the critical edges to have relatively lower stochasticity during training.
> >```
>
>
> On the other hand, our GSINA explicitly constraints the invariant subgraph to be top-r (our $r$ can be as small as 0.1) for sparsity, by defining and solving the OT problem (Eq. 5, 6).
>
>    - Additionally, Gumbel noise used in GSAT is not intended to enhance sparsity. The general purpose of Gumbel noise for machine learning is to enhance “randomness” and introduce perturbations, allowing even small-weight edges to be trained (for both GSAT and GSINA). We have discussed the significance of Gumbel noise in Sec. 2.2.
>
>    - Regarding the results of the ablation study (Table 6), there seems to be a misunderstanding. The observed phenomenon of "GSINA is weaker than GSAT" is because the hyperparameter $r$ chosen for "GSINA without Gumbel" is not optimal. It is directly inherited from the “GSINA with Gumbel” (the complete GSINA model). Therefore, the effectiveness of "GSINA without Gumbel" is not optimal. The key objective of the ablation study in Table 6 is to demonstrate the significance of each component of GSINA (Gumbel noise, node attention). Removing them would result in a drop in GSINA's performance, which is the essence of the ablation study.
>
> **3. Regarding the issue with experiments:**
>
>    - We have already reported “GSAT in the graph-level OOD tasks” in Tab. 2 and 3.
>
>    - For other questions about baselines, the reply to reviewer Jtwt: https://openreview.net/forum?id=MfWFUJklRI&noteId=MGX5WW6hZk might be helpful.
>
> Hope that our responses could address your concerns, if there are any further questions, please feel free to raise.

---

### Official Review · Reviewer_8Epa · 2023-11-01

**Soundness:** 3 good
**Presentation:** 2 fair
**Contribution:** 2 fair
**Rating:** 5
**Confidence:** 4

**Summary:**

This paper proposes a novel graph attention mechanism called Graph Sinkhorn Attention (GSINA) for graph invariant learning (GIL). GSINA extracts sparse, soft, and differentiable invariant subgraphs from input graphs by leveraging the optimal transport theory and the Sinkhorn algorithm. The proposed method acts as a powerful regularization to improve generalization in GIL tasks. The key benefits of GSINA are that it meets the desired principles of sparsity, softness, and differentiability for invariant subgraph extraction. Experiments across synthetic and real-world benchmarks demonstrate that GSINA outperforms prior state-of-the-art GIL methods on both graph-level and node-level tasks.

**Strengths:**

1. Originality: The paper proposes a new graph attention mechanism using optimal transport to extract invariant subgraphs. This is a novel application of the Sinkhorn algorithm not explored before for invariant learning on graphs.

2. Quality: The theoretical analysis explains the design principles and formulations behind the proposed approach. The experiments compare against multiple baselines over several benchmarks to demonstrate the effectiveness of the method.

3. Clarity: The paper is well organized and clearly explains the background, proposed method, and experimental results. Visualizations provide some intuition about the sparse graph attention.

4. Significance: The work introduces a general framework for graph invariant learning that is applicable to both node and graph tasks. It provides improvements over state-of-the-art techniques, showing promise for this approach to invariant learning on graph data.

**Weaknesses:**

1. The paper lacks an in-depth theoretical analysis of why the proposed optimal transport approach and Sinkhorn algorithm can effectively extract invariant subgraphs for graph learning tasks. More analysis connecting GSINA to invariance principles or analyzing its inductive biases could strengthen the method.

2. The paper lacks computational complexity analysis of the proposed GSINA method. It is unclear how the time and space complexity scale as the size of the input graphs increases. Moreover, there is no comparison of the running time or memory usage of GSINA compared to the baseline methods. Analyzing the overhead imposed by using optimal transport and Sinkhorn could quantify the tradeoff between accuracy gains and computational costs.

3. The Introduction section lacks specificity in explaining the innovative contributions of the proposed GSINA model for graph invariant learning. Additionally, the Approach section lacks adequate transition and explanation before formulating invariant subgraph extraction as an optimal transport problem. Addressing these weaknesses would enhance the clarity and logical flow of the paper's core contributions.

**Questions:**

1. The complexity analysis of GSINA is limited in this paper. Can you provide a detailed analysis of the computational overhead of GSINA compared to standard GNNs and other graph invariant learning techniques? What are the asymptotic complexities of key components like Sinkhorn attention and Gumbel trick?

2. This paper argues that sparse, soft, and differentiable are design principles for graph invariant learning (GIL). However, connectivity is another important consideration for extracting meaningful and interpretable invariant subgraphs in GIL. The invariant regions should represent coherent structures and patterns rather than disjoint disconnected components. How do you think about the connectivity of invariant subgraphs? Does the proposed GSINA model take this into account?

---

> ### Author Response · Authors · 2023-11-12
>
> Thank you for your thorough review. In response to your comments, we would like to provide the following supplements and clarifications:
>
> 1. Regarding the theoretical analysis of this paper, **we have added the problem setting of GIL and the inductive bias behind the invariant subgraph method in the background section (moved to Appendix A.1).** For the effectiveness of GSINA in extracting the invariant subgraph, we leverage Mutual Information and formulate the extraction as an optimization problem in Sec. 2.1, which allows GSINA to learn the inference capability for the invariant subgraph from the data. As for how Sinkhorn attention contributes to GIL subgraph extraction, we have analyzed the three properties of GSINA's subgraph in Section 1: sparsity, softness, and differentiability. The first two are important mathematical properties, and the third ensures sound model optimization. In Sec. 2.2, Appendix D.1 (Fig. 11), D.2, and D.3, we have demonstrated and visualized how GSINA accurately extracts an invariant subgraph that satisfies these three properties. Although, as analyzed in https://openreview.net/forum?id=MfWFUJklRI&noteId=LJ14fZcClY , the inference capability of GSINA for the invariant subgraph is not state-of-the-art, it is still showing effectiveness to improve GIL tasks. Besides, existing GIL works such as GSAT and CIGA have already shown that the invariant subgraph could be learned with a GNN-based subgraph extractor, on top of that, the key contribution of our GSINA is to guarantee the sparsity, softness, and differentiability of the extracted invariant subgraph, and our experiments have shown that these considerations are meaningful to improve GIL tasks.
>
> 2. Regarding the computational complexity analysis of GSINA, we have supplemented it in our response to Reviewer mZFa: https://openreview.net/forum?id=MfWFUJklRI&noteId=1nGTCQOItJ . **We have included the analysis in Appendix D.1.** Additionally, as described in Sec. 2.2, the Gumbel trick itself introduces minimal computational burden, consisting of only two steps: 1) sampling Gumbel noise and 2) adding it to the edge score.
>
> 3. We can summarize our contributions as follows: 1) To the best of our knowledge, we are the first to emphasize the importance of sparsity, softness, and differentiability in subgraph extraction for GIL, which was lacking in previous IB and top-k based methods. 2) We propose Graph Sinkhorn Attention (GSINA), a GIL framework that learns the invariant subgraph with controllable sparsity and softness, aiming to improve multiple levels of GIL tasks. 3) Extensive experiments confirm the superiority of GSINA, which consistently outperforms state-of-the-art GIL methods GSAT, CIGA, and EERM by large margins. Besides, the reason for designing the OT problem (Eq. 5, 6) in Sec. 2 is to obtain the invariant subgraph that satisfies the three properties: sparsity, softness, and differentiability, which has been mentioned in Sec. 1 and Sec. 2, and the importance of this OT problem lies in that its solution can satisfy these three requirements in one shot. **We have polished our writing of the introduction and background (Appendix A.1) sections according to your advice.**
>
> 4. Regarding the issue of subgraph connectivity, we believe it depends on the nature of the data. The crucial subgraphs can be either connected or disconnected, for example, a molecule may have multiple disconnected functional groups playing a key role. Connectivity was not explicitly considered in our GSINA design, if desired, a penalty loss (such as the connectivity loss in GIB [1]) can be added to enhance connectivity. However, as we commented in https://openreview.net/forum?id=MfWFUJklRI&noteId=zpwBjxXygV , the subgraph is latent and learned from data, the understanding of data by GNNs does not necessarily have to be consistent with that of humans. Even with no subgraph connectivity guarantee, GSINA can still improve GIL performances.
>
> Thank you again for your valuable feedback. If you have any further questions, please feel free to ask.
>
> [1] Graph Information Bottleneck for Subgraph Recognition.

---

### Official Review · Reviewer_mLys · 2023-11-01

**Soundness:** 2 fair
**Presentation:** 3 good
**Contribution:** 2 fair
**Rating:** 3
**Confidence:** 3

**Summary:**

This paper studies graph invariant learning to discover the invariant relationships between graph data and its labels for different graph learning tasks under various distribution shifts. It adopts the optimal transport theory and designs one graph attention mechanism as a powerful regularization method for graph invariant learning. The experiments show the effectiveness of the method.

**Strengths:**

The strengths of this paper are listed as follows:
- This paper focuses on one important research problem which is graph invariant learning. It is very interesting to me.
- The writing is good in general. The motivations are clearly present. And the technical details are easy to understand.
- The experiments show the improvements on the baselines.

**Weaknesses:**

The concerns are from the following aspects:
- The technical contributions are a little straightforward to me since the key design for graph and graph OOD problem are not very well explained. For example, the design in section 3.1 is similar to GSAT [1] and the edge attention in section 3.2 is very similar to [2]. These differences with existing works are not very clear, which raises my concerns about novelty.
- Some theories are not well formulated. For example, it should formally define the graph distribution by considering the non-Euclidian graph properties. But the theories have weak connections with the graph itself.
- The experiments are not convincing enough since the authors seem to ignore some baselines (such as the graph-level and node-level method in [3]). Some of the results are confusing. For example, why some in-distribution results are lower than the OOD results.

[1] Interpretable and Generalizable Graph Learning via Stochastic Attention Mechanism. ICML 2022.
[2] Debiasing Graph Neural Networks via Learning Disentangled Causal Substructure. NeurIPS 2022.
[3] Out-Of-Distribution Generalization on Graphs: A Survey. ArXiv 2022.

**Questions:**

See weaknesses part above

---

> ### Author Response · Authors · 2023-11-12
>
> Thank you for your review. It has been helpful in improving our writing. We will address your questions point by point.
>
> • **We have added the problem definition in Appendix A.1** to improve the completeness of our paper. And regarding the differences with existing works [1, 2], we would like to provide some necessary clarifications: in regards to the differences with GSAT [1], we have specifically analyzed it in Section 1. GSAT uses the Information Bottleneck to constrain the information of subgraphs, but it has some limitations in its expressiveness. In contrast, we use an OT-based Sinkhorn subgraph extractor (Eq. 5, 6, 7, 8) to constrain the information (i.e. sparsity) of subgraphs. In order for the readers to understand the difference from GSAT, we have already specifically mentioned it in Sec. 2.1. Regarding the differences with the edge attention in Disc [2], as mentioned above, our edge attention is based on the Sinkhorn operation (obtained from Eq. 5, 6, 7, 8). The similarity between our method and Disc, GSAT, CIGA, and other works lies only in Eq. 4, which is the use of GNN to learn edge scores. However, our edge attention is different from edge scores in that we also consider issues of sparsity and softness. Hence, our method has a significant difference compared to Disc. For the technical contribution of this paper, please refer to our response to reviewer 8Epa: https://openreview.net/forum?id=MfWFUJklRI&noteId=zBciutJrKy , **we have summarized our contributions and  supplemented them to Sec. 1.**, our contributions lie in addressing the issues that were present in the previous GIL model, as analyzed in Sec. 1, and resulting in improved performances. We did not extensively describe the problem setting before (but we have supplemented it in Appendix A.1 now) because we did not actually formulate a new graph OOD problem with a new definition. Our problem setting is consistent with other GIL works, such as MoleOOD[4], GIL[5], CIGA, etc.
>
> • The existence of powerful deep learning models like GNN prevents people from specifically focusing on the characteristics of the original input data structure. This has led to a reduced discussion of graph properties in our paper, and instead, a greater focus on the computation and the utilization of GNN. Our GSINA, by optimizing the problem in Sec. 2.1, has modified the message-passing mechanism of GNN in Sec. 2.2 to improve GIL tasks. **The main connection of our method with the graph lies in subgraph extraction**: In Sec. 2.1, we formulated the problem of subgraph extraction using the Variational Mutual Information, and then we defined the subgraph in the form of graph attention in Sec. 2.2. This eventually leads to a greater emphasis on the use of GNN and the optimization of deep learning models, rather than the discussion of graph properties. We have discussed the inductive bias of the invariant subgraph in the background and related works section (Appendix A.1).
>
> • Regarding the issue of lacking baselines, experiment design, and our GSINA's superiority, our response is the same as the one for reviewer Jtwt: https://openreview.net/forum?id=MfWFUJklRI&noteId=MGX5WW6hZk . In it, we discuss the representativeness of the baselines we selected, the differences in their benchmark settings/styles, and the superiority of GSINA compared to them. As for the case of "in-distribution results are lower than the OOD results", it is because the in-distribution testing model is ERM (the same as CIGA setting), not our GSINA model. The advantage of our GSINA model over ERM lies in the ability to perform subgraph extraction. Even for data without distribution shifts, we can still find helpful crucial subgraphs for graph label prediction. This brings an advantage compared to ERM without subgraph extraction in the IID setting.
>
> **We have revised the writing of the introduction and background (in Appendix A.1) sections, and thanks for your advice**, if there are any further questions, please feel free to ask.
>
> [4] Learning substructure invariance for out-of-distribution molecular representations.
>
> [5] Learning invariant graph representations for out-of-distribution generalization.

---

### Official Review · Reviewer_Jtwt · 2023-11-01

**Soundness:** 2 fair
**Presentation:** 3 good
**Contribution:** 2 fair
**Rating:** 3
**Confidence:** 4

**Summary:**

In the context of graph invariant learning, this paper presents the GSINA model, which leverages optimal transport and graph attention to identify invariant subgraphs while satisfying the principles of sparsity, softness, and differentiability. Graph-level and node-level experiments on both synthetic and real-world datasets are carried out to demonstrate the effectiveness of GSINA.

**Strengths:**

(1)	The paper is skillfully written and exhibits a well-structured organization.
(2)	The experiments encompass commonly-used datasets for this task and provide comprehensive comparisons with many typical methods, including CIGA and GSAT. I believe the experiments are robust and information-rich.
(3)	The majority of models relying on top-k selection extract hard subgraphs, often resulting in a notable loss of information in node-level experiments. In contrast, GSINA retains all edges by assigning low attention weights to variant part, ensuring the completeness of the graph structure.

**Weaknesses:**

（1） Using optimal transport theory to obtain differentiable solution to top-k selection has been adopted by other existing works, so I think the method of this paper lacks novelty.

（2） The superiority of GSINA is marginal and not consistent. For example, for Graph-level OOD generalization performances in Figure 4 and Table 5, GSINA expresses worse results.

（3） The compared methods of different datasets are not unified. Baselines in Table2-4 are  less than Table 5.  And they ignored to compare some recently proposed methods: MoleOOD[1],  GIL[2],  Disc[3] and  GREA[4].

（4） The interpretable performance is not enough to verify the effectiveness of GSINA since  the used attention mechanism is expected to have stronger interpretability than other models.

（5） GALA[5] presents GALA for learning invariant graph representations without environment partitions under the proposed minimal assumptions. It is suggested to cite this paper.


[1] Learning substructure invariance for out-of-distribution molecular representations.

[2] Learning invariant graph representations for out-of-distribution generalization.

[3] Debiasing Graph Neural Networks via Learning Disentangled Causal Substructure.

[4] Graph Rationalization with Environment-based Augmentations.

[5] Rethinking Invariant Graph Representation Learning without Environment Partitions

**Questions:**

See Weakness.

**Details Of Ethics Concerns:**

None.

---

> ### Author Response · Authors · 2023-11-11
> **Thank you for your comments. We would like to respond to each of your questions as follows.**
>
> (1) For the novelty, the paper is more of a somewhat novel combination of not-so-novel techniques. Our GSINA incorporates a series of profound works such as the Variational Mutual Information in Sec. 2.1 and the combination of GNN and Sinkhorn Algorithm in Sec. 2.2. By integrating these techniques and considering the task characteristics of subgraph extraction in GIL, we have designed a novel graph attention mechanism in Sec. 2.2 to enhance various levels of GIL tasks. Additionally, our analysis of GIL tasks in Sec. 1 is also innovative, we are the first to highlight the importance of sparsity, softness, and differentiability in subgraph extraction for GIL, which was lacking in previous IB and top-k methods. Besides, the Sinkhorn algorithm has not been explored before for graph invariant learning. All of the above can be regarded as our innovations.
>
> (2) For the superiority of GSINA, perhaps you have misconceptions about our experimental results, and we would like to make necessary clarifications. Fig. 4 in Sec. 4.1 is a hyperparameter sensitivity analysis, where r represents the sparsity or information amount of the subgraphs. The purpose of this experiment is to analyze the impact of the hyperparameter r on the model performance. Fig. 4 does not indicate that GSINA performs poorly. The actual experimental results of the dataset used in Fig. 4 are presented in Table 2, where GSINA outperforms the other approaches consistently. This success can be attributed to the selection of the hyperparameter r based on the validation dataset, as explained in Appendix D.1. Besides, our GSINA performs well on most datasets in Tab.2,3,4,5. As for the results in Table 5, our GSINA achieves the best performance on 5 / 9 datasets, and the performances on the other 4 datasets are comparable to CIGA and much better than other GIL baselines. We have analyzed these results in Sec. 4.1, and for these results, they are hard to improve due to their task difficulties, and most model improvements are "marginal" as well. As CIGA models are also SOTA, it is reasonable to claim the superiority of GSINA for GIL.
>
> Furthermore, we have found that increasing the patience of early stopping to 20 (previously 10) is beneficial to reducing underfitting on the Drug-Assay/Sca datasets. Compared to before, we have achieved greater improvements, and now they outperform other baselines consistently, and there is only a small gap compared to the strongest CIGAv2.
>
> (3) The datasets are not unified for a fair comparison, as GSAT and CIGA experimental settings are very different (in datasets, GNN backbones, etc.), and they both have a lot of hyperparameter tuning work, which resulted in difficulty in merging these two different styles of benchmarks together in the experiments. Therefore, we conducted separate experiments, which effectively proved that our approach is indeed more effective than our selected baselines.
>
> There are some baselines [1,2,3,4] that we did not compare with, and samely, they use various datasets and GNN backbones, making it challenging to unify the comparison. Additionally, GSAT, CIGA, and EERM, which we primarily compare with, are highly representative and have higher citation counts than [1,2,3,4]. However, for some datasets used in [1,2,3,4] that have the same experimental settings as ours (such as MoleOOD's Drug, GIL's Spmotif, and Graph-SST2), we can directly compare their reported results to demonstrate our superiority, as shown in the table below:
>
> ||Spmotif-0.5|Spmotif-0.7|Spmotif-0.9 | | Graph-SST2|| Drug-Assay| Drug-Sca | Drug-Size|
> |-|-|-|-|-|-|-|-|-|-|
> | MoleOOD|/|/|/||/||71.38 ± 0.68|68.02 ± 0.55|66.51 ± 0.55|
> |GIL|54.56±3.02| 53.12±2.18 | 46.04±3.51|| 83.44±0.37 ||/|/|/|
> | GSINA (GIN) | **55.16±5.69** |**56.83±6.32**|**49.86±6.10**| | **83.66±0.37** | | **72.53±0.54** | **69.07±0.47** | **67.48±0.33** |
>
> It can be seen that our GSINA has superiority compared to MoleOOD and GIL. It is necessary to emphasize the use of datasets for these baselines, with the most special being Disc, which uses CMNIST-75sp, CFashion-75sp, and CKuzushiji-75sp for experiments, is totally different from others; in addition, MoleOOD, GIL, and GREA focused on testing the Ogbg dataset (such as testing various GNN backbones or pooling functions), but our GSINA, along with our main comparison of GSAT and CIGA, did not pay enough attention to Ogbg. GSINA and GSAT only had the Ogbg experimental results of PNA backbones, which were not aligned with these baselines, so we did not report them.
>
> (4) We discuss the issue of interpretability in Appendix D.2 that GSINA, as a top-k based method, is naturally weak in interpretability. The attention mechanism used does not necessarily imply better interpretability, hence we also do not claim it as a contribution. Our focus is on the generalization ability of GSINA.
>
> (5) We are pleased to cite this paper to enrich the content of our paper.
>
> Please let us know if you have any further questions or concerns.

---

> ### Author Response · Authors · 2023-11-13
> **Weakness 4: Further Discussions of Interpretability**
>
> To address weakness 4, which is the issue of interpretability in GSINA, we would like to provide further explanations. As observed in the results shown in Tables 11 and 12, the interpretability of GSINA is weaker compared to GSAT, and we believe this can be explained by the following reasons:
>
> 1) As analyzed in Appendix D.2, the use of top-k operand is not suitable for interpretability metrics such as AUC-ROC and precision. This is because top-k only focuses on whether the result is 0 or 1, without considering the ranking relationship between the data.
>
> 2) Due to the utilization of soft top-r operation in GSINA, it places more mathematical constraints on the graph attention value distribution compared to GSAT. The attention of GSAT is more flexible because it calculates the probability of each edge belonging to the invariant part separately, without considering the global constraint on the ratio (r) of the invariant subgraph. The choice of r in GSINA significantly impacts the attention distribution, whereas GSAT is not subject to this issue, which is why GSAT's attention distribution is more flexible.
>
> 3) The design of GSAT, as mentioned in 2) and the second point in https://openreview.net/forum?id=MfWFUJklRI&noteId=zpwBjxXygV , is a tradeoff. It relaxes the constraint for subgraph sparsity and gains more interpretability, but it falls short in effectively (sparsely) filtering out variant parts (as described in Section 1, Figure 1). This leads to GSAT having a less optimal ability to make predictions using subgraphs compared to GSINA.
>
> **We have incorporated the above analysis into Appendix D.2 to enrich the content of the article.** Very thankful for your advice.

---

### Official Review · Reviewer_mZFa · 2023-11-03

**Soundness:** 3 good
**Presentation:** 2 fair
**Contribution:** 3 good
**Rating:** 5
**Confidence:** 3

**Summary:**

This paper proposes a new method (GSINA) for learning invariant representations when performing node or graph classification. The method is designed to provide sparse invariant features (subgraphs) like simultaneously ensuring differentiability. They draw inspiration from attention based strategies, which are differentiable, and top-k strategies, which are typically sparse but not differentiable. GSINA combines these two perspectives and uses proposes an optimal transport problem to obtain edge importances. The found invariant subgraph is then feed into the predictor to learn more generalizable features. The authors perform comprehensive experiments to demonstrate the efficacy of their method.

**Strengths:**

- The authors perform a very comprehensive evaluation across many datasets and tasks. Indeed, the proposed method does substantially outperform its competitors. For example, on by almost 10% with respect to CIGA on Table 4.

- The proposed method appears to be well-grounded. Maximizing the mutual information between a subgraph and the label is a popular approach in both the invariant feature learning and GNN-explanation literature. Moreover, the use of Optimal Transport on top of attention makes sense given the shortcomings of existing approaches.

- Their method does not seem to require extensive hyperparameter tuning (just r), which uses the validation performance. This is especially helpful for OOD settings, where one cannot assume access to OOD data.

**Weaknesses:**

- I think the novelty is a bit lacking in the approach. Individual pieces seem like combinations of existing pieces. Perhaps I missed something and the authors could clarify this? Could the authors also clarify if the softmasks learnt by GSINA do in fact coincide with the known invariant signals in the graph (as per the explanation literature)?

- Runtime/ Computational Complexity: I'm concerned that this method will substantially increase the runtime relative to the vanilla model and also other invariant representation methods. Could the others please provide some runtime plots so that I can understand if this is in fact the case?

- The writing of this paper needs to be polished. For example, "The invariance optimization methods are based on the principle of invariance, which assumes the invariant property inside data, i.e. the invariant features under distribution shifts."

**Questions:**

Please see the weakness above, and below.

Out of curiosity, I was wondering if GSINA could be applied to a pretrained GNN as a way of post-hoc improving the representations for better OOD generalization? Perhaps by enforcing consistency between the pretrained model's predictions on the original graph and the extract subgraph or through end-to-end fine tuning?

Also, can the authors also please clarify if GSINA + the GNN Predictor are trained "end to end?" They mention a two-stage framework, but I just wanted this clarified. Maybe adding a pytorch-style algorithm would be beneficial.

---

> ### Author Response · Authors · 2023-11-11
>
> Thanks for your comments.
>
> Firstly, with regard to the novelty of the paper, the paper is more of a somewhat novel combination of not-so-novel techniques. Our GSINA incorporates a series of profound works such as the Variational Mutual Information in Sec. 2.1 and the combination of GNN and Sinkhorn Algorithm in Sec. 2.2. By integrating these techniques and considering the task characteristics of subgraph extraction in GIL, we have designed a novel graph attention mechanism in Sec. 2.2 to enhance various levels of GIL tasks. Additionally, our analysis of GIL tasks in Sec. 1 is also innovative. To the best of our knowledge, we are the first to highlight the importance of sparsity, softness, and differentiability in subgraph extraction for GIL, which was lacking in previous IB and top-k based methods. Besides, the Sinkhorn algorithm has not been explored before for invariant learning on graphs. All of the above can be regarded as our innovations. For issues of novelty and contribution, also refer to https://openreview.net/forum?id=MfWFUJklRI&noteId=MGX5WW6hZk point 1, and https://openreview.net/forum?id=MfWFUJklRI&noteId=HaGQUZvbQ1 point 1.
>
> Secondly, regarding the interpretability of GSINA's node and edge weights, we have conducted experiments in Appendix D.2, and D.3 and provided interpretability metrics and visualizations. We have also compared them with GSAT and performed an analysis. Our experimental conclusion is that GSINA does not achieve state-of-the-art performance in interpretability compared to GSAT. We tend to explain this phenomenon as a natural drawback of top-k based GIL approaches (discussed in Sec. 1). While our GSINA is a top-k based GIL approach, GSAT is not. Intuitively, the top-k operation is not sensitive to the relative order of elements (a requirement for interpretability) but only cares about whether it is 0 or 1. However, our main focus is on the accuracy of classification problems, and in that aspect, the effectiveness of our GSINA is at the state-of-the-art level. For more discussions about interpretability, please see https://openreview.net/forum?id=MfWFUJklRI&noteId=LJ14fZcClY .
>
> Thirdly, regarding computational complexity, overall, our GSINA shares the same two-stage architecture (subgraph extraction and prediction based on the subgraph) with GSAT and CIGA, here is no more complexity. For the OT problem (Eq. 5), it can be solved with a few iterations of element-wise tensor computations (Eq. 6). The number of Sinkhorn iterations (Eq. 6) could be rather small, and in this paper, we set it to 10 globally, which is not an expensive choice. **We have added runtime/complexity analysis in Appendix D.1 to improve our paper.**
>
> We have polished our writing, thanks for your advice.
>
> Regarding Question 1, unfortunately, we believe that combining GSINA with pretraining is challenging. In the implementation of GSINA (Sec. 2.2), we use two independent GNNs in the G -> G_S -> Y pipeline, each responsible for the computation of G -> G_S and G_S -> Y. In the G -> G_S process, we obtain attention weights for each node and edge in the graph structure. In the G_S -> Y process, we utilize the obtained attention weights to modify the message-passing mechanism for representation learning. If we are given a pretrained GNN of G -> Y, we find that its task lacks similarity with the tasks of G -> G_S and G_S -> Y. Therefore, we have not yet figured out how to utilize such a pretrained GNN. Essentially, GSINA can also be seen as modifying the internal structure of the GNN for G -> Y.
>
> Regarding Question 2, the answer is yes. We emphasized the importance of GIL’s differentiability in Sec. 1. Our two-stage G -> G_S -> Y pipeline allows gradient propagation and forms an end-to-end process. **We have added the specific algorithm in Appendix A.4 to enrich the content of the paper.**
>
> If you have any further questions, please feel free to ask.

---

> ### Author Response · Authors · 2023-11-12
> **Runtime / Computational Complexity**
>
> We conducted experiments on the training time of the models on all datasets of the CIGA benchmark (Statistics of the benchmark datasets can be found in Appendix B, Table 9). We used the following code snippet to record the mean ± std training time (in terms of ms/epoch):
>
> ```python
>
> start = torch.cuda.Event(enable_time=True)
> end = torch.cuda.Event(enable_time=True)
>
> start.record()
>
> # train an epoch
>
> end.record()
> torch.cuda.synchronize()
>
> elapsed = start.elapsed_time(end) # in millisecond
>
> ```
>
> Our GSINA has two different settings for subgraph extraction in a batch. The first (denoted as micro) setting is to compute soft top-r for each graph in the batch, which is slower. The second (denoted as macro) setting is to compute soft top-r just once for the entire batch (which can be considered as a large graph composed of several smaller graphs), which is faster. The macro setting computes the top-r for the whole graph, relaxing the constraint of top-r on each individual graph, which is useful when the distribution shifts are complicated and difficult to find a reasonable $r$ for all graphs. Therefore, the model selection between micro and macro is dependent on the respective characteristics of each dataset, we used micro on Spmotif, TU-PROT, TU-DD, and macro on Graph-SST5, Twitter, Drug-Assay, Drug-Sca, Drug-Size, TU-NCI1, TU-NCI109.
> On the other hand, CIGA uses hard top-k selection for subgraph extraction for each graph (similar to the micro setting) in the batch. And we have implemented GSAT (without hyperparameter tuning) on the CIGA benchmark for training time comparison, GSAT is similar to the macro setting, GSAT models the probability $p$ of each edge belonging to the invariant part, which does not consider how many graphs are in the batch.
>
> We compare the training time of the following models: 1. not GIL, ERM , 2. GIL, GSAT, 3. GIL, CIGA, 4. GIL, GSINA (-macro), and 5. GIL, GSINA (-micro):
>
> |             | Spmotif                      | Graph-SST5                   | Twitter                      | Drug-Assay                     | Drug-Sca                      | Drug-Size                      | TU-NCI1                      | TU-NCI109                   | TU-PROT                     | TU-DD                        |
> | ----------- | ---------------------------- | ---------------------------- | ---------------------------- | ------------------------------ | ----------------------------- | ------------------------------ | ---------------------------- | --------------------------- | --------------------------- | ---------------------------- |
> | ERM         | 5797.0464 ± 215.3673         | 3993.3391 ± 99.6189          | 2123.6215 ± 289.1412         | 9771.7190 ± 2122.2238          | 6704.2010 ± 658.8927          | 10814.6467 ± 1104.6025         | 1258.8495 ± 71.3337          | 1217.0766 ± 77.9733         | 400.4288 ± 80.5055          | 369.0103 ± 92.8588           |
> | GSAT        | 11084.8443 ± 290.5553        | 7737.8888 ± 84.9996          | 4129.1175 ± 138.3962         | 17026.1180 ± 657.9675          | 10845.3604 ± 597.7060         | 17952.6367 ± 1071.1815         | 2450.3426 ± 98.2116          | 2394.2303 ± 81.2618         | 714.7085 ± 79.7937          | 717.2253 ± 89.7630           |
> | GSINA-macro | 14799.7564 ± 315.1995        | 9503.5158 ± 234.1883| 5214.8427 ± 125.0164| 19833.7992 ± 1240.6085 | 12675.0961 ± 670.9245| 23152.3664 ± 1540.6074| 3141.5663 ± 116.0735 | 2995.5683 ± 90.5246| 891.0935 ± 74.9539 | 1035.9521 ± 63.5705          |
> | CIGA        | 21658.0957 ± 908.5341        | 14344.7820 ± 246.2677        | 7958.9061 ± 208.1967         | 54833.6117 ± 1221.3473         | 33741.7516 ± 615.7502         | 58453.0594 ± 924.8912          | 5014.3386 ± 126.6357         | 4481.9476 ± 54.5312         | 1190.0512 ± 82.4018         | 1361.7259 ± 113.0401         |
> | GSINA-micro | 34576.2656 ± 628.3021| 24521.2875 ± 369.2448        | 13038.0312 ± 214.8203        | 97431.7109 ± 3121.8496         | 59474.7531 ± 689.3121         | 101284.0609 ± 751.5329         | 7429.4210 ± 113.3016| 7373.4569 ± 125.3487        | 1929.6349 ± 94.4660 | 2141.5195 ± 109.6949|
>
>
> In the experimental results, ERM is always the fastest, and our second setting (GSINA-macro) is often much faster than CIGA and a bit slower than GSAT due to iterative computations (Eq. 6); while the first setting (GSINA-micro), also due to iterative computations (Eq. 6) in subgraph extraction, is slower than CIGA's hard top-k selection. However, since we can set the number of iterations to a relatively small value (we uniformly used 10), GSINA does not introduce a significant computing overhead and is always within 2 x CIGA's complexity.
>
> **We have included the above analysis in Appendix D.1: Model Selection to enhance the completeness of the paper.** Thank you for raising the question.